# Uniform spatial pooling explains topographic organization and deviation from receptive-field scale invariance in primate V1

Y. Chen[1,2,3], H. Ko[1,3], B. V. Zemelman[2,4], E. Seidemann[1,2,3] & I. Nauhaus [1,2,3]✉

Receptive field (RF) size and preferred spatial frequency (SF) vary greatly across the primary visual cortex (V1), increasing in a scale invariant fashion with eccentricity. Recent studies reveal that preferred SF also forms a fine-scale periodic map. A fundamental open question is how local variability in preferred SF is tied to the overall spatial RF. Here, we use two-photon imaging to simultaneously measure maps of RF size, phase selectivity, SF bandwidth, and orientation bandwidth—all of which were found to be topographically organized and correlate with preferred SF. Each of these newly characterized inter-map relationships strongly deviate from scale invariance, yet reveal a common motif—they are all accounted for by a model with uniform spatial pooling from scale invariant inputs. Our results and model provide novel and quantitative understanding of the output from V1 to downstream circuits.

[1] Department of Psychology, University of Texas, Austin, TX, USA. [2] Department of Neuroscience, University of Texas, Austin, TX, USA. [3] Center for Perceptual Systems, University of Texas, Austin, TX, USA. [4] Center for Learning and Memory, University of Texas, Austin, TX, USA. ✉email: nauhaus@utexas.edu

Primary visual cortex (V1) is responsive to image features at widely varying spatial scales, which can be quantified by measuring each neuron's receptive field (RF) size and spatial frequency (SF) tuning. Studies have shown that variation in spatial tuning exists at both a global and local level of organization across the V1 sheet. The global organization of RF size and SF tuning has been characterized with relatively coarse electrode sampling—as one moves further from the foveal representation, RFs scale up in degrees of scene coverage to account for a reduction in sampling density (i.e. magnification factor)[1–3]. Modeling this global trend of RFs as univariate scaling, or "scale invariant", aligns with the graded resolution of inputs from the retina[4–6]. Next, studies also show substantial local variation in SF tuning for a given eccentricity. More specifically, local variability of preferred SF is organized into periodic clustering, with a spatial period that matches orientation preference[7–14], thus placing it inside the V1 "hypercolumn"[2] of about 1 mm. Local maps of preferred SF can be presumed to ride on top of the more global eccentricity-dependent gradient of preferred SF. The question remains as to whether V1 RFs scale in proportion to preferred SF, within the hypercolumn, thus predicting other locally periodic maps of RF size and SF bandwidth. A general model of the spatial RF within the macaque V1 hypercolumn, which can only be constrained by dense sampling of neuronal responses at this local scale, is needed to identify V1's coverage of the visual scene at each location[2,15,16].

Given the accuracy of scale invariance at describing the global architecture, along with its pervasiveness as a V1 model in general, it serves as a valuable reference point in characterizing spatial RFs in the local architecture. If scale invariance holds locally, then the periodic preferred SF maps[12] predict the mapping of other scale parameters, such as SF bandwidth and RF size, but not others, such as orientation selectivity and phase selectivity. Some electrode studies that limit recordings to a range of eccentricities suggest that local V1 maps are not scale invariant. For instance, within parafoveal V1, spatial scale parameters are not always proportional[10,17–21], and features of orientation tuning and SF tuning are often correlated[21,22]. Importantly, these studies reveal that at a minimum, constrained modeling of V1's local deviation from scale invariance requires measurement of both spectral (e.g. Fourier) and spatial domains of the RF, along with identification of "simple" and "complex" populations[17,23,24]. At the same time, we still lack a comprehensive model of deviations from scale invariance within the hypercolumn, which requires densely sampled measurements.

We performed two-photon imaging of the genetically-encoded calcium indicator GCaMP6f[25–27] in anesthetized macaque V1 to simultaneously measure preferred SF, RF width, SF bandwidth, orientation bandwidth, and phase selectivity. All of the measured parameters showed significant clustering at close cortical distances. Scale invariance predicts that the relationship between preferred SF maps and the other parameters is either independent (viz. orientation and phase selectivity) or proportional (viz. RF width and SF bandwidth), yet this was a poor predictor of the data in every case. Next, we showed that the architecture in superficial V1 can instead be described using a model that integrates over a population of scale invariant RFs, which we refer to as "pooled scale invariance". Our results and model provide a quantitative account, and deeper understanding, of the organization of V1 and its outputs to extrastriate cortex.

## Results
### Defining the model of scale invariant V1 RFs. A persistent theme of this study is comparing superficial V1 tuning to models of RF scale invariance. In general, the term "scale invariance"

indicates that shape does not change with scale. Therefore, a scale invariant model of V1 RFs implies that the aspect ratio (length/width) of the RF boundary is constant, as is the number of ON/OFF subfields inside the RF boundary; e.g. see the two cartoon RFs along the solid blue diagonal line in Fig. 1a. Constraints imposed by scale invariance on V1 RFs can be conveniently formalized in a Gabor model. First, the width of the Gaussian envelope, $\sigma_{x,\mathrm{si}}$ (°), is inversely proportional to the sine wave carrier frequency, $f_o$ (cyc/°): $\sigma_{x,\mathrm{si}}(f_o) = 1/(\pi f_o)$. We also define $f_o$ as the "preferred" or "peak" SF of a neuron's SF tuning curve. This constraint states that there is one cycle of $f_o$ inside $\pi \sigma_{x,\mathrm{si}}$. The use of $\pi$ as the proportionality constant is based on previous studies of primate and cat V1 simple cells[18,28–30], which indicates 2–3 ON–OFF subfields. $\pi$ is also consistent with a difference-of-Gaussians (where $\sigma_1/\sigma_2 = 2$) or second derivative-of-Gaussian, which are commonly used models of V1 RFs. In turn, all

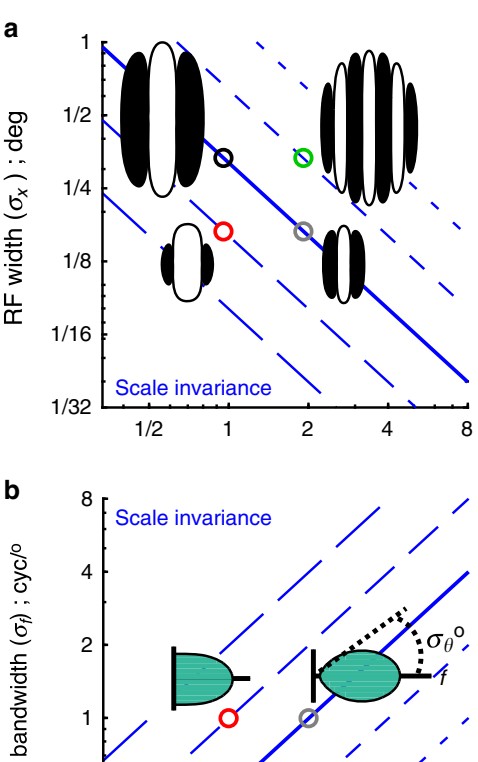

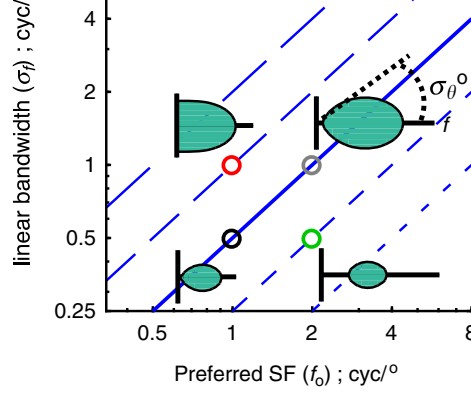

**Fig. 1 Illustration of scale invariance. a** RF width (*y*-axis) is inversely proportional to $f_o$ (*x*-axis) in a model of scale invariance. Points that are common to any of the five blue diagonal lines are part of a scale invariant family. Four example Gabor's are shown at the four locations indicated by open circles. Prior studies have typically modeled V1 RFs as sitting along the solid blue line (Eq. (1)). **b** SF bandwidth (*y*-axis) is proportional to $f_o$ (*x*-axis) in a model of scale invariance. Each location represents a 2D Gaussian that is constrained by the Fourier transform of the Gabors in **a**. Each blue line in **b** is calculated from the Fourier transform of the family of Gabor's along each line of **a**. Also, the 2D Fourier transform of the four Gabors shown in **a** are overlaid, where the *x*-axis (cyc/°) is along the preferred orientation. The green ellipse indicates the distribution of 2D spectral energy. Along a line of scale invariance (e.g. solid blue), orientation bandwidth ("$\sigma_\theta$") top right) does not change.

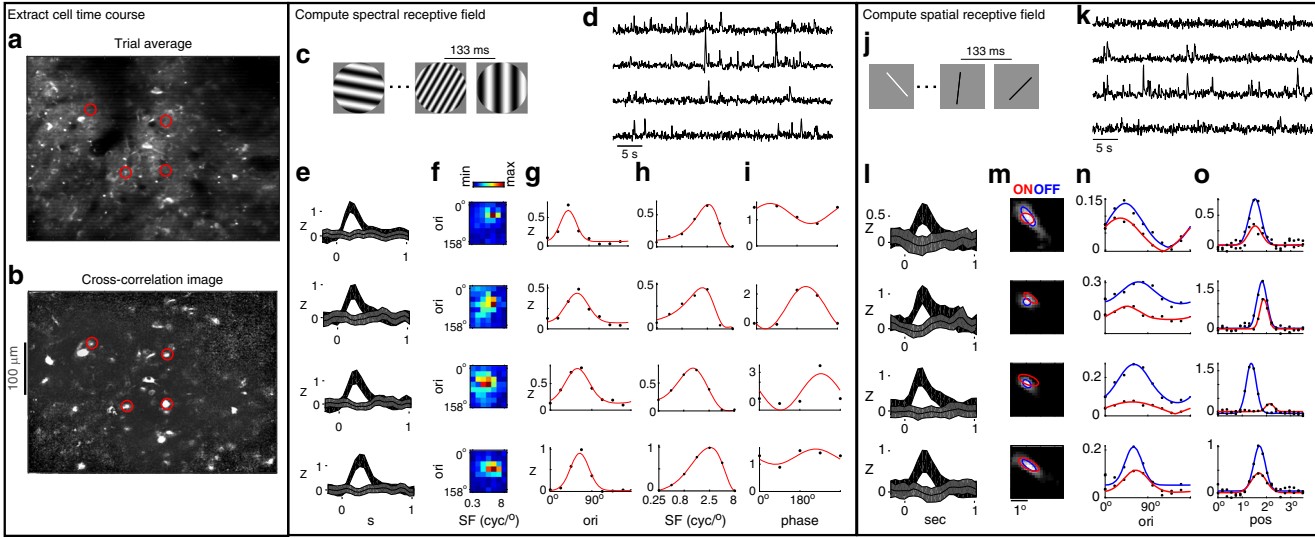

**Fig. 2 Measuring spectral and spatial RFs with two-photon calcium imaging of GCaMP6f. a** Average image from the first trial (40 s). **b** The local cross-correlation image was used to identify groups of pixels that have common temporal dynamics. Four examples of manually selected neurons are outlined in red. Their mean responses are shown in panels to the right. **c** Illustration of the random grating stimulus. **d** Time courses from one trial of the four example cells in **a**. **e** Mean and SE of response following stimulus onset. The two traces in each panel correspond to the orientation/SF combination that elicited the largest (black) and smallest (gray) response. **f** Color map indicating the normalized response to each orientation and SF in the ensemble. Responses were averaged over spatial phase. The color range spans the minimum (blue) to maximum (red) response of each cell. **g** Orientation tuning curve (black dots) computed by taking a weighted sum over the SF dimension. Gaussian fit is in red. **h** Same as **g**, but for SF. Fit is a difference-of-Gaussian. **i** Spatial phase tuning curve for the orientation and SF that elicited the largest mean response. The fit is a sine wave. **j** Illustration of the random bar stimulus. **k** Time courses from one trial of the four example cells in **a**. **l** Mean and SE response after onset of the bar that elicits the largest (black) and smallest (gray) response. **m** Grayscale image is the addition of the ON and OFF spatial RF; i.e. the "spatial envelope". The half-max contour of the ON and OFF 2D Gaussian fits are shown in red and blue, respectively. **n** The ON and OFF orientation tuning curves (black), measured by averaging over the position dimension, with Gaussian fits (red and blue). **o** The ON and OFF line weighting functions (black) taken at each cell's optimal orientation, with Gaussian fits (red and blue). This experiment was performed once in three different regions-of-interest, in two animals.

subsequent derivations and results are placed in the context of $\sigma_{x,\text{si}}(f_o) = 1/(\pi f_o)$ ("Methods", Eq. (1)), which is the solid blue line in Fig. 1a. Other models of scale invariance that use coefficients different than $\pi$ are also shown (Fig. 1a, dashed blue).

Taking the Fourier transform of scale invariant Gabors in the spatial domain (°) yields tuning in the spectral domain (cyc/°) that is also scale invariant. In general, scale invariance in the spectral domain implies proportionality between $f_o$ and SF bandwidth (cyc/°), and a constant orientation bandwidth (Fig. 1b). The Fourier transform of Gabors along the solid blue line in Fig. 1a gives 2D Gaussians along the solid blue line in Fig. 1b. Along the radial dimension of each 2D Gaussian is the SF tuning, which is constrained by $\sigma_{f,\text{si}} = f_o/2$, where $\sigma_{f,\text{si}}$ is the width of the Gaussian (cyc/°) and $f_o$ is the center (cyc/°). The aspect ratio (length/width) of the 2D Gaussian energy in the spectral domain does not vary with $f_o$ under scale invariance, thus yielding constant orientation bandwidth.

The illustrations in Fig. 1a, and the above formulations, are described in the context of the class of V1 RFs known as "simple cells". Simple cells have relatively non-overlapping ON and OFF subfields[17], and are thus more linear. These same formulations apply to the "energy model" of V1 "complex cells", in which the output of multiple linear RFs, which vary only in phase, are squared and summed[31]. Although basic and tractable, testing the model of scale invariance within the hypercolumn requires that all the aforementioned parameters be measured simultaneously from a densely sampled population, which we have done here with two-photon imaging.

**Measuring spectral and spatial tuning curves in upper L2/3.** We performed two-photon calcium imaging of excitatory

neurons in primate V1, after expression of GCaMP6f[25] with AAV1-CaMKII. Virus injections and chamber maintenance prior to recordings were identical to the procedures described in ref. [27], with the exception that we implanted a novel chamber design suitable for chronic two-photon imaging (Fig. S9). To extract the time courses of neurons in the ROI, we computed the local cross-correlation image during stimulus presentation, followed by manual selection of bright puncta in the ROI (Fig. 2a, b) (see "Methods" for details). We present results from two visual stimuli that were shown to the same population of cells. One stimulus was a rapid sequence of static sine wave gratings (Fig. 2c), and the other stimulus was a rapid sequence of narrow bars (Fig. 2j). The flashed gratings were used to quantify tuning for orientation, SF, and spatial phase (Fig. 1e–i)[32], whereas the random bar stimulus yielded tuning for orientation, location, and size of the RFs (Fig. 1l–o)[33]. Together, the two stimulus sets allow for a characterization of simple or complex V1 neurons. For more linear (i.e. simple) cells, only one of the two stimuli are necessary to make similar measurements, but many of the cells in the recorded population are complex, based on the separation of ON and OFF components (Fig. 2o) and the depth of phase modulation (Fig. 2i). Table 1 provides summary statistics of the tuning parameters measured in the three imaging regions in this study.

**Pooling scale invariant RFs over retinotopy accounts for RF width.** Simple cell RFs in V1 are classically depicted as a Gabor with about one cycle of $f_o$ inside the envelope (Fig. 1a, solid blue line), with the same constraint generally applied to the linear components of complex cells[31]. We begin with a descriptive comparison between the RF width and $f_o$ to highlight how their statistics deviate from scale invariance. First, most RFs are wider than the scale-invariance

prediction (Fig. 3a, blue line) of $1/(\pi f_o)$. At the same time, cells with the lowest $f_o$ (largest predicted width) are near the scale invariance prediction. Next, the standard deviation of $\log_2(f_o)$ (Fig. 3a, x-axis) is over 2.2-fold greater than the standard deviation of $\log_2$(RF width) (Fig. 3a, y-axis) ($p < 10^{-20}$; F-test), whereas these should be equal under scale invariance. Finally, a linear fit to the data in log–log coordinates has a shallow slope of $-0.08$ ($r = -0.18$; $p = 0.025$) (Table 2, column 1), whereas scale invariance predicts a clearer trend with a slope of $-1.0$. These descriptive inconsistencies with scale invariance form the basis of the model described below.

We modeled the measured RF widths in our two-photon data as the result of pooling of scale invariant RFs. This is expressed as

$$\sigma_{x,p}^2 = \sigma_{x,si}^2(f_o) + \sigma_{h(x)}^2 \quad \text{RF width (°) model} \quad (5)$$

where $\sigma_{x,p}$ is the observed RF width at the output, $\sigma_{x,si}(f_o) = 1/(\pi f_o)$ is the RF width of scale invariant inputs from Eq. (1) (Fig. 1a, solid blue line), and $\sigma_{h(x)}$ is the width of the Gaussian function that

weights the pool of scale invariant inputs, all in "degrees of visual field". Note that the model assumes convolution between the scale invariant RFs and the Gaussian pooling function to produce the measured RF widths, in which case the variances of the convolved functions add. In turn, we refer to this as "pooled scale invariance". The known variables from the data are $\sigma_{x,p}$ and $f_o$, which allowed us to estimate $\sigma_{h(x)} = 0.24°$ using the entire data set, giving the green line in Fig. 3a. This parameter estimate was similar when measured independently for the 3 ROIs: $\sigma_{h(x)} = [0.18°\ 0.24°\ 0.24°]$. A simulation of scale invariant pooling at $f_o = 2$ cyc/° is illustrated in Fig. 3c, showing the population of scale invariant inputs (blue) and the RF envelope at the output (green). The pooling model has a similar correlation with the data as a linear fit in log–log coordinates (Table 2), but uses only one parameter. Furthermore, the pooling width, $\sigma_{h(x)} = 0.24°$, provides a physical interpretation of pooling within the cortical space (mm). Multiplying $\sigma_{h(x)}$ by the magnification factor (mm/°) gives an estimate of the lateral spatial integration in millimeters of cortex. Magnification factor was measured by first calculating the cortical separation (mm) and RF separation (°) between all cell pairs in the three ROIs, followed by fitting a line through the origin of the scatter plot of mm vs. deg (°), which yielded a slope of 2.0 mm/°. RF separation was measured between the peaks of the RF envelopes, determined from the 2D Gaussian fits (Fig. 2m). In all, 2 mm/° is comparable to previous studies that recorded near a similar eccentricity ($\sim2°$)[3,33,34], and predicts integration from up to about 0.48 mm away ($1\sigma$), which could be attributed to both local and more long-range unmyelinated inputs in superficial V1 (ref. [35]).

The two panels in Fig. 3d show the maps of RF width based on the prediction of the two models under study, scale invariance (bottom) and pooled scale invariance (top). Both are a function of $f_o$, which exhibits significant pairwise clustering (Fig. 3f, blue). However, the predicted map of RF width from pooled scale invariance, along with the actual map in Fig. 3e, are relatively flat. Nevertheless, the pairwise statistics show that neuron pairs are

**Table 1 Marginal statistics.**

|  | $f_o$ | $\sigma_x$ | $\sigma_f$ | $\sigma_{\log(x)}$ | $\sigma_\theta$ | F1/F0 | $\frac{\lvert\mu_{ON}-\mu_{OFF}\rvert}{\sigma_{ON}+\sigma_{OFF}}$ |
|---|---|---|---|---|---|---|---|
| Mean | 2.12 | 0.3 | 1.43 | 0.81 | 26.01 | 1.01 | 0.2 |
| Median | 2.06 | 0.29 | 1.38 | 0.73 | 24.12 | 0.90 | 0.16 |
| SD | 0.93 | 0.06 | 0.38 | 0.25 | 10.58 | 0.72 | 0.17 |
| Mean-$\log_2$ | 0.95 | −1.77 | 0.47 | – | 4.60 | −0.43 | −3.01 |
| SD-$\log_2$ | 0.64 | 0.29 | 0.38 | – | 0.52 | 1.27 | 1.65 |

This table provides standard statistics of the main RF properties in the data set under study. The symbols on top represent each RF property. From left to right, they are preferred SF (cyc/°), RF width (°), linear SF bandwidth (cyc/°), logarithmic SF bandwidth (octaves), orientation bandwidth (°), spatial phase selectivity, and ON–OFF separation. Each row is a different statistic computed across the three ROIs: mean, median, standard deviation (SD), mean of the $\log_2$, and SD of the $\log_2$. The SD-$\log_2$ metrics in the bottom two rows are particularly useful in comparing the distribution between parameters that are presumed proportional under scale invariance. For example, the SD of $\log_2$(RF width) is equal to the SD of $\log_2(f_o)$ under scale invariance, regardless of the scaling factor used, but there is a more than twofold difference (F-test; $p < 10^{-20}$). Similarly, the SD of $\log_2$(linear SF bandwidth) is narrower than the SD of $\log_2$(SF preference) (F-test; $p < 10^{-10}$).

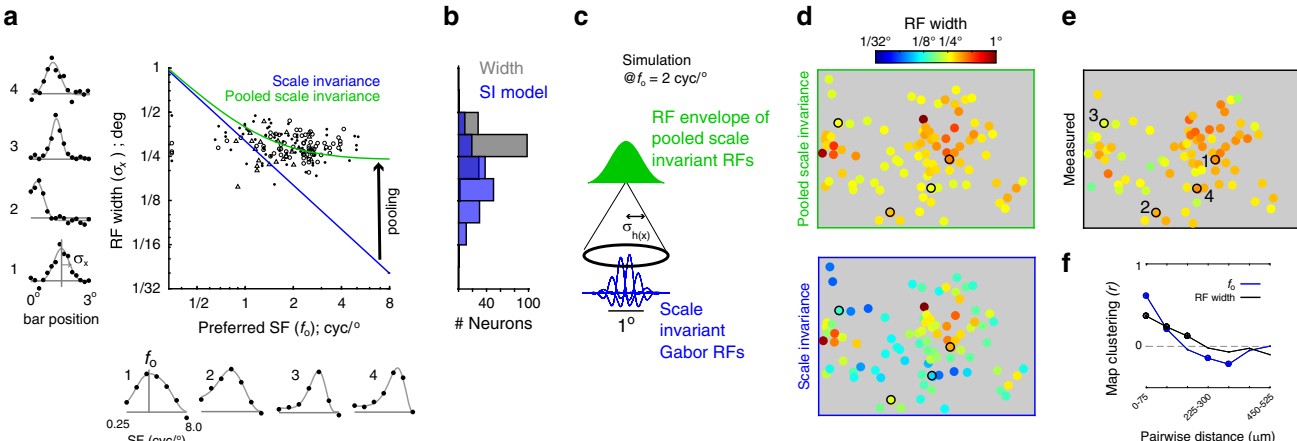

**Fig. 3 Receptive field width as a function of preferred SF ($f_o$). a** The scatter plot compares $f_o$ to RF width from three ROIs, each indicated by a different symbol. See Table 2 for joint statistics. The five data points at the extreme left of the x-axis were "low pass" (see "Methods"). To the left of the y-axis are position tuning curves (black dots) and Gaussian fits (gray) of four example neurons in one ROI outlined and enumerated in **e**. Below the x-axis are the SF tuning curves and fits of the same four cells. **b** Distribution of measured RF width and the scale invariance prediction of RF width based on $f_o$. **c** Illustration of the pooling model in the spatial domain, at $f_o = 2$ cyc/°. At bottom are the 1D Gabor functions from the model of scale invariance. They are shifted and weighted according to the Gaussian in the pooling model ($\sigma_{h(x)} = 0.24°$). The superposition of the Gabors' energy (i.e. their envelopes) yields the wider green Gaussian on top, which is the output of the pooling model. **d** Bottom and top panels are the predicted map of RF width by plugging $f_o$ into the scale invariance model and pooled scale invariance model, respectively. The yield in these map are based on tuning fits to the random grating stimulus (see "Methods"). **e** Map of the measured RF width. The yield in **d** is different from **e** because they are based on tuning fits from two separate stimulus blocks, the random gratings and random bars (Fig. 2c, j), respectively. Like the map in **e**, the scatter plot in **a** showed the intersection of the yield between the two different stimulus blocks. **f** Pearson correlation coefficient between neuron pairs, within 75 μm cortical distance bins, using all three ROIs in the study. The symbol at each distance bin indicates the correlation coefficient's significance: Closed dot is $p > 0.01$, asterisk is $p < 0.01$, asterisk and open circle is $p < 0.001$.

**Table 2 Joint statistics.**

|  | $\log_2(1/\sigma_x)$ | $\log_2(\sigma_f)$ | $\sigma_{\log(f)}$ | $\log_2(\sigma_\theta)$ | F1/F0 | $\frac{|\mu_{ON}-\mu_{OFF}|}{\sigma_{ON}+\sigma_{OFF}}$ |
|---|---|---|---|---|---|---|
| **$\log_2(f_o)$** |  |  |  |  |  |  |
| \<difference\> | 0.84 | −0.48 | −0.15 | 3.65 | 0.06 | −0.76 |
| $r$ | 0.18 | 0.50 | −0.76 | −0.55 | −0.51 | −0.22 |
| $p$ | 0.025 | 2.6e−12 | 3.6e−34 | 2.1e−15 | 6.0e−13 | 0.004 |
| Slope | 0.08 | 0.29 | −0.31 | −0.45 | −0.59 | −0.06 |
| Intercept | 1.69 | 0.19 | 1.1 | 5.03 | 1.58 | 0.25 |
| **$\log_2[P(f_o)]$** |  |  |  |  |  |  |
| $r$ (SI pooling) | 0.21 | 0.52 | 0.8 | 0.54 | 0.46 | N/A |
| $p$ (SI pooling) | 0.01 | 3.1e−13 | 8.4e−41 | 1.2e−14 | 2.1e−10 | N/A |

Top 5 rows: Joint statistics between the $\log_2(f_o)$ and other variables in this study. The first four column symbols—$\sigma_x$, $\sigma_f$, $\sigma_{\log(f)}$, $\sigma_\theta$—are RF width (°), linear SF bandwidth (c/°), logarithmic SF bandwidth (octaves), and orientation bandwidth (°). The last two column symbols are spatial phase selectivity and ON–OFF overlap. As indicated, $\log_2$ was taken for parameters in columns 1, 2, and 4, prior to comparing with $\log_2(f_o)$. The first row is the average difference between the column variable and $\log_2(f_o)$. The second and third rows are the Pearson correlation coefficient and $p$ value relating the column variable to $\log_2(f_o)$. The fourth and fifth rows are the slope and intercept of the linear fit, where $\log_2(f_o)$ is the domain. Bottom 2 rows: The first five column parameters have a prediction from the pooled scale invariance model, which is a function of $f_o$, generically identified as $P(f_o)$ on the left. Specifically, the prediction of the variables in columns 1–5 are given by Eqs. (5), (7)–(9) and (6), respectively. The Pearson correlation coefficient and $p$ value between the model predictions and data are given in the bottom two rows.

more likely than chance to have similar RF width at short cortical distances (Fig. 3f, black). In summary, there is a shallow and noisy relationship between RF size and $f_o$, which is consistent with a mapping of RF size that is more weakly clustered than the map of $f_o$.

The rectangular bar in our visual stimulus is composed of Fourier energy that decays at higher SFs. Therefore, it has the potential to create $f_o$-dependent errors in measurements of RF width. The Supplementary Information addresses specific issues along these lines. First, the bar might be expected to drive a subset of the population more than others, which could bias the RF width at the output of a static nonlinearity. However, the analyses in the Supplementary Section IV.3 show that most of the neurons are theoretically and empirically well-driven by the 0.2° bars without significant dependence on their SF tuning. This suggests that nonlinearities are unlikely to have a major effect on RF size vs. $f_o$. Second, Supplementary Section VI quantifies the widening induced by the finite bar width, as it could inflate RF width and the estimate of the pooling window. A simple deconvolution derivation shows that it has minimal impact on the calculation of the 0.24° pooling window, and therefore the Fig. 3 results in general.

**Pooling scale invariant RFs over retinotopy accounts for phase selectivity.** Early studies of V1 RFs segregated the V1 population into two groups, "simple cells" and "complex cells", based on the amount of spatial overlap between ON and OFF responses[17]. Later, the distinction was quantified as phase selectivity, specifically "F1/F0", at a neuron's $f_o$[23]. Simple cells have larger F1/F0 (strong $f_o$ phase selectivity) whereas complex cells have smaller F1/F0 (weak $f_o$ phase selectivity). We assessed F1/F0 as a function of each neuron's $f_o$ (Fig. 4a) and deviation from scale invariance (Fig. 4b). Figure 4a shows that neurons with lower $\log_2(f_o)$ have greater F1/F0 ($r = -0.51$; $p < 10^{-12}$), which is consistent with the previous result that simple cells have lower $f_o$ than complex cells[21]. Figure 4b shows that neurons with an RF width near the scale invariance prediction (i.e. near unity on the $x$-axis) have greater F1/F0 ($r = -0.51$; $p < 10^{-11}$). Consistently, it has been shown that complex cells, unlike simple cells, tend to have much larger RF width than predicted by a period of $f_o$[10,17,18]. Next, we show that these results can be accounted for by the model of pooled scale invariance.

In the "Methods", we used pooled scale invariance to formulate F1/F0 as a function of $f_o$, which is the following Gaussian:

$$F1(f_o)/F0 = \pi * \exp[-(\sigma_{h(x)}2\pi f_o D)^2/2] \quad \text{phase selectivity model.}$$
(6)

The SD is $(\sigma_{h(x)}2\pi D)^{-1}$, where $\sigma_{h(x)} = 0.24°$ is the pooling window that was computed to account for RF width in the previous section. $D$ is a free parameter that scales the rate of absolute phase progression within the pooling window. $D = 0$ and 1 correspond to constant "absolute" and "relative" phase, respectively (Fig. 4a, dashed green lines). The absolute phase is referenced by the fovea, whereas the relative phase is referenced by the center of the RF envelope. A least-squares fit to the entire data set yields $D = 0.54$ (Fig. 4a, solid green), indicating that the absolute phase is more clustered than a model whereby the population maintains the constant relative phase. This parameter estimate was similar when measured independently for the 3 ROIs: $D = [0.55\ 0.56\ 0.58]$. From this model, we derived the result that complex cells tend to have wider RFs than simple cells (Fig. 4b green). Specifically, the model domain in Fig. 4b is the ratio between two predicted RF widths—pooled scale invariance (Eq. (1)) and scale invariance (Eq. (5)). The domain of the data, however, is the ratio between measured RF width and the scale invariance prediction. Unity on the $x$-axis of Fig. 4b corresponds to a standard Gabor model with 2–3 subfields, and higher values indicate additional subfields within the envelope.

Finally, the link between $f_o$ and F1/F0 predicts a previously unidentified map of F1/F0 (Fig. 4d), which we found by comparing neuronal pairs at varying distance. Like $f_o$, nearby neurons were more likely than chance to have similar F1/F0 (Fig. 4f). To summarize, for a given pooling window in visual degrees ($\sigma_{h(x)}$), scale invariant inputs with higher $f_o$ will have a broader distribution of absolute spatial phase, resulting in a summed output with a relatively large drop in phase selectivity.

**Pooling scale invariant RFs in the spectral domain accounts for SF bandwidth.** In the preceding section, we showed that most superficial neurons in V1 have substantially wider RFs than predicted by classic models of scale invariance—i.e. wider than one period of $f_o$. Before proceeding to the following analyses on SF bandwidth and orientation bandwidth, we first emphasize that widening of the RF via pooling in the spatial domain (°) does not necessarily lead to a narrowing of tuning in the spectral domain (c/°). Given the inverse scaling properties of the Fourier transform, this may be a tempting conclusion. However, inverse scaling assumes (1) linear superposition of sinewaves in the spatial domain and (2) power in the spectral domain is built from the constructive interference of sinewave components. However, the preceding section describes pooled scale invariance as a hierarchical model where a majority of cells at the output have weak phase selectivity due to pooling from variable phase. The

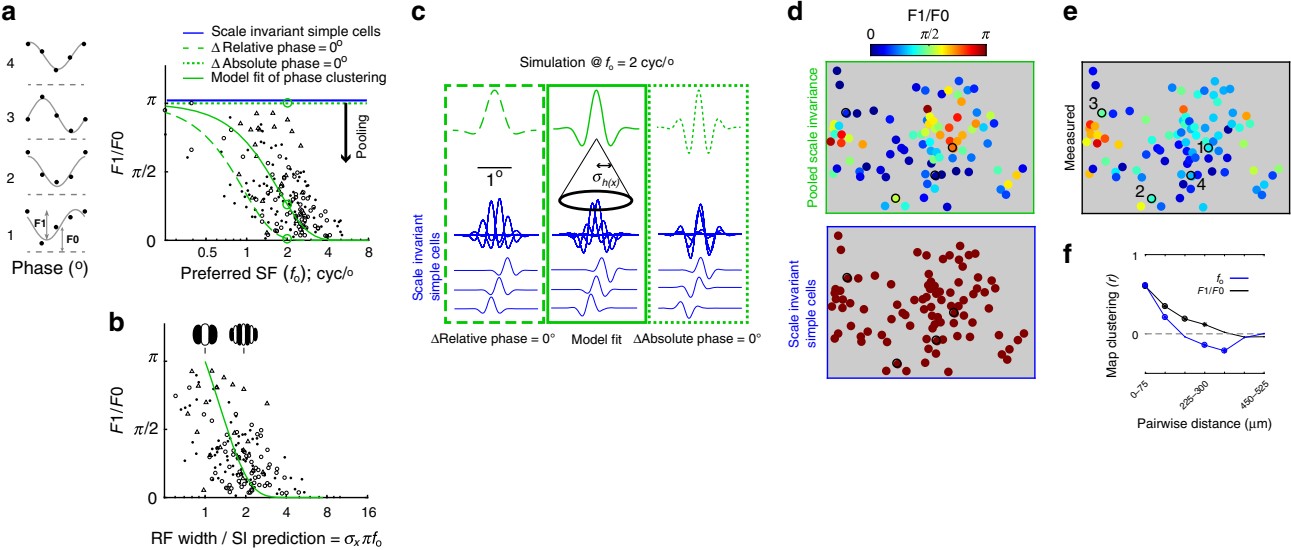

**Fig. 4 Phase selectivity as a function of preferred SF ($f_o$). a** Scatter plot compares $f_o$ to $F1/F0$ from three ROIs, each indicated by a different symbol. See Table 2 for joint statistics. Each of the three green curves show the result of pooling from the same population of scale invariant simple cells, and the same retinotopic integration window ($\sigma_{h(x)} = 0.24°$; determined in Fig. 3), but with different alignment in the absolute phase. The solid green curve is the least-squares fit of phase alignment to the data. To the left of the y-axis are phase tuning curves (black) and fits (gray) of four example neurons in one ROI outlined and enumerated in **e**. **b** Like the scatter plot in **a**, y-axis data points are $F1/F0$. However, the x-axis is converted into a metric for the deviation from scale invariance. Specifically, it is the measured RF width divided by the scale invariance prediction of RF width. To obtain the domain of the green curve, we took the following ratio: pooled scale invariance prediction of RF width (Eq. (5)) over the scale invariance prediction of RF width (Eq. (1)). Gabor insets illustrate how the number of ON/OFF subfields increase along the x-axis. **c** Simulation of the pooling model in the spatial domain at $f_o = 2$ cyc/°, for the three examples of phase alignment plotted in **a** (see green open circles in **a**). At bottom in blue are the 1D Gabor functions from the model of scale invariance. The spatial phase progresses at a different rate, inside each of the three green rectangles. In the left example, "relative phase" does not change, whereas in the right example "absolute phase" does not change. Just above, also in blue, are the Gabor functions weighted according to the Gaussian in the pooling model ($\sigma_{h(x)} = 0.24°$). The top green curves are the superposition of scale invariant inputs, where the constant absolute phase (right) yields the greatest phase modulation. **d** Maps of $F1/F0$. Bottom panel is all simple cells ($F1/F0 = \pi$) and is the input to the scale invariant pooling model. Top panel is the output of scale invariant pooling, based on the solid green fit to the data in **a**. **e** Measured map of $F1/F0$. **f** Pearson correlation coefficient between neuronal pairs, within 75 μm cortical distance bins, using all three ROIs in the study. The symbol at each distance bin indicates the correlation coefficient's significance: Closed dot is $p > 0.01$, asterisk is $p < 0.01$, asterisk and open circle is $p < 0.001$.

output neurons nonetheless accumulate "energy" in the spectral domain due to a rectification of the inputs. Responsiveness to sinewave gratings are calculated as the average over the spatial phase (i.e. $F0$), not $F1$ amplitude, in which case the effects of pooling in one domain cannot be uniquely derived from pooling in the other.

As shown in Fig. 1b, SF bandwidth (cyc/°) is proportional to $f_o$ under scale invariance. The axes of the Fig. 1b illustration are identical to the axes of the Fig. 5a scatter plot, and has the same solid blue line showing the specific model of scale invariance used here, $\sigma_{f,si} = f_o/2$. We measured $\sigma_{f,si}$ as the half-width at 61% of the peak in the DoG fit. The data are more consistent with *pooled* scale invariance, for the same reasons described above for RF width (Fig. 3). For one, the measured widths are consistently wider than the scale invariance prediction. Second, the slope of a linear fit ($r = 0.50$; $p < 10^{-11}$) in log–log coordinates is 0.29, whereas the slope in any model of scale invariance is 1.0. Lastly, the standard deviation of log(SF bandwidth) is 59% as large as log($f_o$) (F-test, $p < 10^{-10}$) (Table 1; row 5), yet they are predicted to be equal under scale invariance. In summary, the deviations of RF width and SF bandwidth from the scale invariance prediction are analogous. Both can be explained as a widening that is most pronounced when the underlying scale invariant predictions are most narrow. We therefore applied the same scale invariant pooling model in the spectral domain to model SF bandwidth. In the case of RF width, the deviation (widening) occurs at higher $f_o$. In the case of SF bandwidth, the deviation occurs at lower $f_o$.

To predict the measured SF tuning, the pooled scale invariance model uses a weighted sum of Gaussian functions that are constrained by scale invariance, $\sigma_{f,si} = f_o/2$ (Eq. (2)). The weighting function is a Gaussian in the SF domain, and has a constant width, $\sigma_{h(f)}$. In turn, SF bandwidth can be estimated by

$$\sigma_{f,p}^2 = \sigma_{f,si}^2(f_o) + \sigma_{h(f)}^2 \quad \text{linear SF bandwidth (c/°) model}, \quad (7)$$

where $\sigma_{f,p}^2$ and $f_o$ are measured, so the one free parameter is $\sigma_{h(f)}$. We estimated $\sigma_{h(f)}$ to be 0.85 cyc/°, giving the green line in Fig. 5a. This estimate of $\sigma_{h(f)}$ from the entire data set was similar to the estimates from each of the three independent ROIs: $\sigma_{h(f)} = [0.75\ 0.87\ 0.85$ cyc/°]. The pooling model has similar correlation to the data as a linear fit (Table 2), but uses only one free parameter. Furthermore, $\sigma_{h(f)}$ can be related to pooling within the functional architecture. However, relating $\sigma_{h(f)}$ to integration in the $f_o$ maps is not as straightforward as the retinotopy (see previous section) because the $f_o$ maps are not locally monotonic, but periodic. In turn, we cannot convert $\sigma_{h(f)}$ into microns of cortex using a simple scale factor. However, the standard deviation of $f_o$ across all three ROIs was 0.93 cyc/°, which is just above the estimate of $\sigma_{h(f)}$ at 0.85 cyc/°. This requires that the diameter ($2\sigma$) of the pooling window in microns of cortex be slightly smaller than the spatial period of the SF maps ($\sim$750 μm)[12], which is comparable to the estimate of the cortical window estimate of 960 μm ($2\sigma$) using RF width, described above.

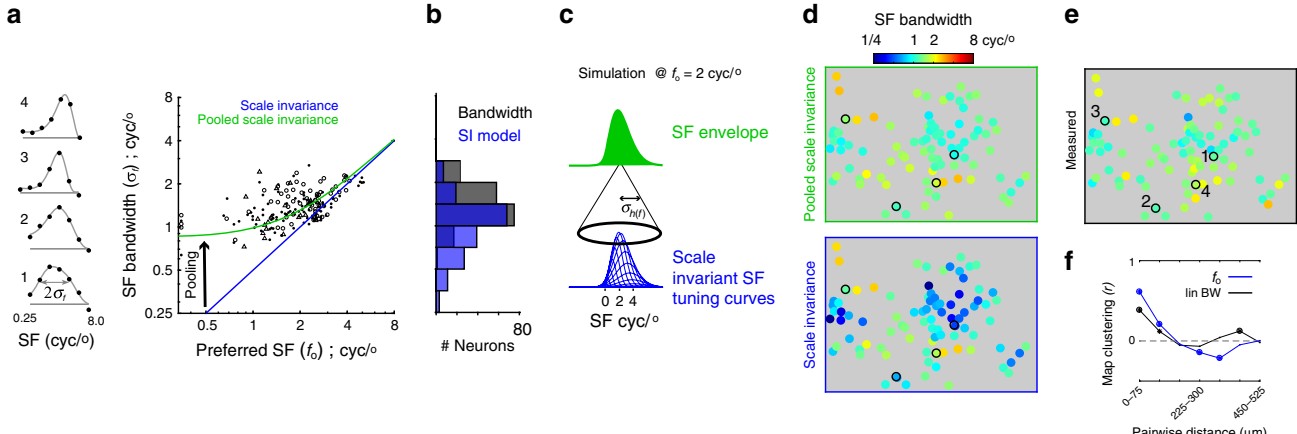

**Fig. 5 SF bandwidth as a function of preferred SF ($f_o$). a** The scatter plot compares $f_o$ to SF bandwidth in three ROIs, each indicated by a different symbol. See Table 2 for joint statistics. The five data points at the extreme left of the $x$-axis were low pass. Blue line is the scale invariance prediction of bandwidth, $f_o/2$. Green line is the model of pooled scale invariance (Eq. (7)). To the left of the $y$-axis are SF tuning curves (black dots) and fits (gray) of four example cells in one ROI outlined and enumerated in **e**. **b** Distribution of the measured SF bandwidth (black) and the scale invariance prediction of SF bandwidth (blue). **c** Simulation of the pooling model in the spectral domain, at $f_o = 2$ cyc/°. At the bottom are the 1D Gaussian functions from the model of scale invariance. They are shifted (cyc/°) and weighted according to the Gaussian in the pooling model ($\sigma_{h(f)} = 0.85$ cyc/°). The superposition of the scale invariant Gaussians yields the function on top, which is the output of the pooling model. **d** Predicted maps of SF bandwidth. Bottom and top panels are generated by plugging $f_o$ in to the scale invariance and pooled scale invariance model, respectively. **e** Map of measured SF bandwidth. **f** Pearson correlation coefficient between neural pairs, within 75 μm cortical distance bins, using all three ROIs in the study. The symbol at each distance bin indicates the correlation coefficient's significance: Closed dot is $p > 0.01$, asterisk is $p < 0.01$, asterisk and open circle is $p < 0.001$.

SF bandwidth in the linear domain (cyc/°) was modeled above, as opposed to bandwidth in the logarithmic domain (octaves), because it allowed for a simpler description of the data's deviation from scale invariance. That is, a single invariant pooling mechanism in the linear SF domain (i.e. constant $\sigma_{h(f)}$) creates preferential widening of bandwidth at lower $f_o$, which is not the case for invariant pooling in the logarithmic domain. Furthermore, pooling in the linear SF domain is more amenable to deriving the effects of scale invariant pooling on orientation bandwidth, described in the next section. However, logarithmic bandwidth in octaves is the more commonly used metric for quantifying SF tuning selectivity[21]. Figure S3 shows how the same pooling model can account for the dependency of log bandwidth (Eq. (8)) on $f_o$[21]. The derivation of the pooling model from linear bandwidth to logarithmic bandwidth is described in the "Methods" section (Eq. (8)).

Next, we characterized the functional architecture of linear SF bandwidth. A map is expected given its significant dependence on $f_o$. Indeed, there is significant clustering at close distances (Fig. 5f, black), yet it is weaker than the scale invariance prediction, which is consistent with a reduced range of bandwidth imposed by pooled scale invariance. The marginal and joint statistics of SF bandwidth (linear and logarithmic) and $f_o$ are summarized in Tables 1 and 2.

**Pooling scale invariant RFs in the spectral domain accounts for orientation bandwidth**. Under scale invariance, orientation bandwidth ($\sigma_\theta$) is independent from $f_o$ (Fig. 1b). However, Fig. 6a shows a significant negative correlation between $\log(\sigma_\theta)$ and $\log(f_o)$: ($r = -0.55$; $p < 10^{-14}$; slope/intercept = $-0.45/5.03$). We are not aware of this correlation being reported elsewhere, yet it is consistent with the combined observations in prior studies that report a positive correlation between orientation bandwidth and log SF bandwidth, and a negative correlation between log SF bandwidth and $f_o$[21,22]. Importantly, the trend is also predicted by the model of pooled scale invariance when pooling is extended to the orthogonal dimension of spectral coordinates—orientation and SF are polar dimensions in 2D spectral coordinates (Fig. 6c).

A given pooling window at lower $f_o$ (nearer the origin) pools from a broader distribution of orientation preferences than the same pooling window at higher $f_o$, thus yielding broader tuning at the output of the model. A simulation to illustrate the dependence of orientation tuning on pooling in 2D spectral coordinates, as a function of $f_o$, is shown in Fig. 6c. The pooling window is a circularly symmetric Gaussian with $\sigma = 0.85$ cyc/°, which was the parameter fit in the pooling model for SF bandwidth in the previous section (Fig. 5a, green). Next, the simulation was formalized analytically. We derived the dependence of orientation bandwidth on $f_o$ (Fig. 6a, green), based on pooling from a scale invariant population in the Fourier plane, given as Eq. (9) in the "Methods". No additional parameters were fit to arrive at the model of orientation bandwidth vs. $f_o$; however, it required an estimation of the aspect ratio of the scale invariant RF envelopes (length/width); an aspect ratio of 2.0 was plugged into Eq. (9), based on previous studies[28-30]. The pooling model prediction of orientation bandwidth is similarly correlated to the data as a linear fit (Table 2), yet provides a more parsimonious description of the data, as it uses the same pooling coefficient that was used to predict SF bandwidth from $f_o$.

Due to the strong correlation between orientation bandwidth and $f_o$, there is an expected mapping of orientation bandwidth. The pooling model of orientation bandwidth and the data have maps that are visually comparable (Fig. 6d, e). Like SF bandwidth and RF size, the clustering of orientation bandwidth is weaker than, yet similar in structure to, $f_o$ (Fig. 6f). As discussed later, this relationship between maps of orientation bandwidth and $f_o$ falls in line with previous studies on tuning within V1 maps.

**Summary of the pooled scale invariance model and its performance**. The model of pooled scale invariance uses each cell's $f_o$ to predict four RF properties—RF width, $F1/F0$, SF bandwidth, and orientation bandwidth. The complete model applied to one ROI is shown in Fig. 7. At the bottom of the hierarchy are scale invariant simple cells (Fig. 7b), which have a RF width (°) and SF bandwidth (cyc/°) that are determined by $f_o$ (Fig. 7a). To determine RF width and $F1/F0$ at the output stage (Fig. 7c), the pooling

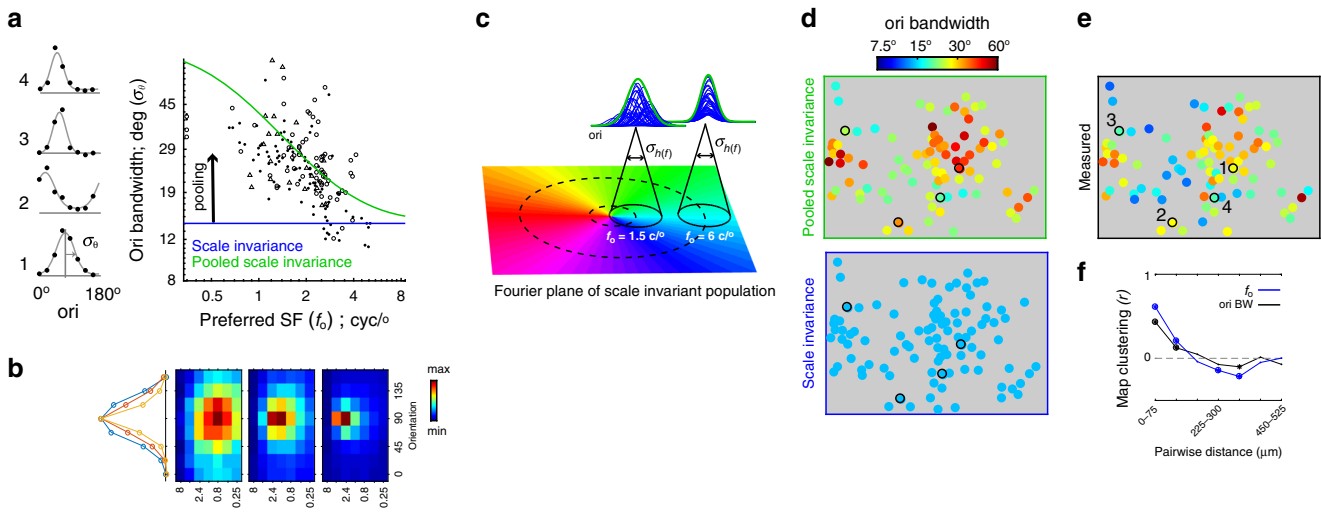

**Fig. 6 Orientation (ori) bandwidth as a function of preferred SF ($f_o$). a** The scatter plot compares $f_o$ to ori bandwidth in three ROIs, each indicated by a different symbol. See Table 2 for joint statistics. The five data points at the extreme left of the x-axis were low pass in SF. The blue line is the ori bandwidth based on a scale invariant aspect ratio of 2.0. The green line is the model of pooled scale invariance, which did not require any additional parameters. Rather, it was derived from the fit to SF bandwidth (Fig. 5a, green). To the left of the y-axis are ori tuning curves (black dots) and Gaussian fits (gray) of four example cells in one ROI outlined and enumerated in **e**. **b** Each of the three images represents the average response from a different subpopulation, as a function of SF (x-axis) and ori (y-axis). Blue-to-red represents normalized response from min-to-max, for each panel. The image on the right is the mean over cells with $f_o > 2$ cyc/°, the middle image is over cells with $1 < f_o < 2$ cyc/°, and the left image is over $f_o < 1$ cyc/°. Curves on the left are the mean ori tuning curves computed from the three images on the right; they are averaged over the SF dimension. Blue, red, and yellow are for the images from left to right, respectively. **c** Simulation of the pooling model in the 2D spectral domain, at $f_o = 1.5$ and 6 cyc/°. At bottom is an angled perspective of the 2D Fourier plane, where the color represents orientation and $f_o$ increases linearly away from the center. Each location of the 2D Fourier plane corresponds to a different 2D scale invariant Gaussian. Pooling closer to the origin (@ 1.5 cyc/°) yields broader ori tuning than pooling further away from the origin (@6 cyc/°), assuming an invariant pooling function ($\sigma_{h(f)} = 0.85$ cyc/°). The simulation on top shows a random sample of weighted and shifted ori tuning curves (blue) from the two indicated pooling locations, along with their superposition (green). **d** Modeled maps of ori bandwidth. Bottom and top panels are generated by plugging $f_o$ into the scale invariance and pooled scale invariance model, respectively. **e** Measured maps of ori bandwidth. **f** Pearson correlation coefficient between neural pairs, within 75 µm cortical distance bins, using all three ROIs in the study. The symbol at each distance bin indicates the correlation coefficient's significance: Closed dot is $p > 0.01$, asterisk is $p < 0.01$, asterisk and open circle is $p < 0.001$.

model integrates over the retinotopy using a weighted Gaussian with $\sigma = 0.24°$, which translates to $\sigma = 0.48$ mm of cortex based on the magnification factor (°/mm). To arrive at the estimate of $F1/F0$, an additional parameter constrains the pooling model, which describes the absolute phase progression, relative to the retinotopic progression, $D$. Finally, a third parameter is used by the pooling model to describe the bandwidth of both orientation and SF in the 2D spectral domain. Pooling in the spectral domain is analogous to pooling in the spatial domain—it is the integration of scale invariant RFs. However, the domain is in cyc/°, and the weighting function is a 2D Gaussian with $\sigma = 0.85$ cyc/°. The layer of pooled scale invariance (Fig. 7c) is visibly more similar to the data (Fig. 7d) than the layer of scale invariant RFs (Fig. 7b).

Next, we quantified the accuracy of pooled scale invariance, relative to scale invariance. All predicted values were generated using "leave-one-out cross-validation", whereby the training population used to predict a given data point consisted of all other data points. Pooled scale invariance generates significantly less mean-squared error (MSE) than scale invariance in predicting $\log_2$(RF width) (paired t-test; $p < 10^{-17}$; 95% CI = [0.77 1.15]) and $\log_2$(SF bandwidth) (paired t-test; $p < 10^{-12}$; 95% CI = [0.34 0.56]). For a more conservative comparison to scale invariance, we then fit "scaling coefficients" that predict RF width and SF bandwidth from $f_o$. That is, we did not predict RF width from $1/(\pi f_o)$, but fit a coefficient other than $\pi$ in this equation. Similarly, we did not assume SF bandwidth = $f_o/2$, but fit a separate scaling coefficient other than ½. The dashed blue lines in Fig. 1 show other examples of scale invariance models that are

allowed in the fitting procedure. Even with the added flexibility to scale invariance, pooled scale invariance has significantly lower MSE in predicting $\log_2$(RF width) (paired t-test; $p < 10^{-12}$; 95% CI = [0.34 0.57]) and $\log_2$(SF bandwidth) (paired t-test; $p < 10^{-9}$; 95% CI = [0.17 0.32]). In the case of $F1/F0$ and orientation bandwidth, the scale invariant model does not have an independent variable, so a comparison of errors was not done. However, there is a significant correlation coefficient between the predictions of pooled scale invariance and the data for both $F1/F0$ and orientation bandwidth (Table 2, row 7, columns 4 and 5).

## Discussion

Preferred SF ($f_o$) varies at a local and global scale of the V1 architecture. At the global scale, across eccentricity, $f_o$ varies with other RF properties in a scale invariant fashion. At the local scale, within the hypercolumn, these relationships were previously untested with sufficient experimental or analytical precision to model V1 output. We offer a model that is a simple hierarchical step beyond scale invariance that provides a far better account of our two-photon imaging data. In Figs. 3–6, each scatter plot compared $f_o$ to another classic V1 tuning parameter from a localized population. Pooled scale invariance performs much better than scale invariance at predicting RF width and SF bandwidth from $f_o$. Furthermore, unlike scale invariance, the pooled scale invariance model is able to use $f_o$ to account for variability in orientation bandwidth and phase selectivity. Given the clustering of $f_o$ within macaque V1, pooled scale invariance

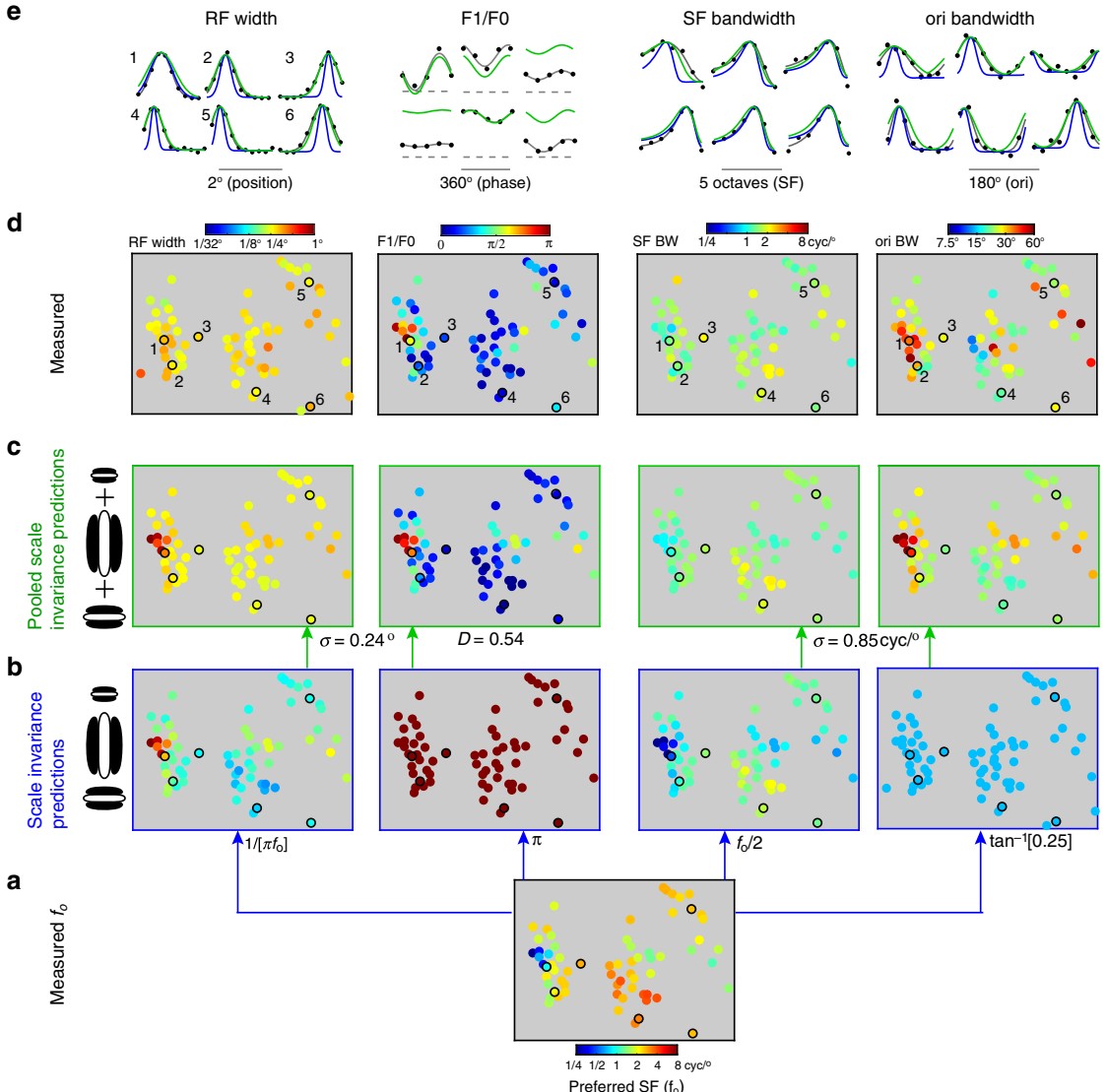

**Fig. 7 Summary of pooled scale invariance model and its performance relative to scale invariance.** Each of the four columns in **b** through **e** corresponds to a different tuning parameter that is predicted by scale invariant pooling, labeled on top. All functional maps shown in this figure are from the same ROI. Data yield is the same across maps, with the exception of the measured RF width at top left, which is due to it being measured with a different visual stimulus and inclusion criterion. **a** Bottom row shows the measured map of $f_o$, which is the independent variable in the models of scale invariance and pooled scale invariance. **b** First layer of the scale invariant pooling model consists of scale invariant simple cells. Below each map is the formula defining the given parameter. From left to right the maps are of RF width ($1\sigma$), F1/F0, SF bandwidth ($1\sigma$), and orientation bandwidth ($1\sigma$). **c** Output layer of the scale invariant pooling model. Under each map is the relevant parameter value(s) required to make the predicted map shown. **d** Measured functional maps shown above their respective model predictions. **e** Measured tuning and model fits of six neurons from this ROI. A number is adjacent to each tuning curve that indicates the neuron in the labeled map below in **d**. Black dots are data, gray line is the direct fit to the data, blue line is the scale invariance prediction, and green line is the pooled scale invariance prediction. In the case of the F1/F0 tuning curves, the dashed gray line is the baseline.

predicts clustering of all the other spatial tuning maps in this study, which was indeed observed. However, maps of RF position (i.e. retinotopy), $f_o$, and preferred orientation had the strongest clustering (Figs. S1 and S2), consistent with them being the domains of pooling in spatial (Eq. (5)) and spectral coordinates (Eq. (7)). Some of the results described here can be gleaned from previous studies that relate single cell tuning to the surrounding architecture[8,36–38], along with studies showing deviation from scale invariance[10,17,18,21,22], yet their hitherto disconnection made it difficult to provide a simple and holistic model of V1 output.

**Predicting orientation bandwidth from preferred SF can be linked to previous results.** We observed a strong correlation

between orientation bandwidth and $f_o$, which we accounted for with isotropic pooling of scale invariant RFs in the 2D spectral domain. To our knowledge, this specific result has not been reported, yet is predictable from previous electrophysiology studies. A positive correlation was shown between logarithmic SF bandwidth and orientation bandwidth, along with a negative correlation between logarithmic SF bandwidth and $f_o$[21,22]. Taken together, this predicts a negative correlation between orientation bandwidth and $f_o$, like in Fig. 6a.

The negative correlation between orientation bandwidth and $f_o$ predicted by pooled scale invariance is also consistent with a putative alignment between orientation pinwheels and regions of low $f_o$ in the functional architecture. This logic is based on the following series of observations. First, V1 neurons have broader

orientation tuning in more diverse regions of the orientation map (e.g. near pinwheel centers)[37,39,40]. Next, pinwheel centers align with the centers of ocular dominance columns[13,41,42], and the centers of ocular dominance columns align with regions of low $f_o$[8,33]. Together, regions of broadly distributed orientation preferences (pinwheel centers) are more likely to align with low $f_o$, as illustrated in Fig. 6b. In turn, isotropic pooling within the maps can be expected to integrate a wider range of orientations, at lower $f_o$ (Fig. 6c), thus producing a negative correlation between orientation bandwidth and $f_o$ (Fig. 6a). This may also be a source of noise in the use of $f_o$ to predict orientation bandwidth in the scale invariant pooling model—there will be randomness in the alignment between local minima in the maps of $f_o$ and regions of diverse orientation preference.

**RF size depends on the visual stimulus used**. V1 RF size is stimulus dependent, so a single value is unable to characterize each neuron. Measurements of RF size are often classified as either a minimum response field (mrf) or a spatial summation field (ssf). mrf is measured from sparse bars or dots that are flashed over the RF[43], whereas the ssf is often measured from drifting gratings of variable size that are centered on the RF[44–46]. The relationship between mrf and ssf has not been well characterized in the same population, but reports of parafoveal mrf[2,3,47,48] tend to be smaller (~0.2–1.0°) than ssf (~1.0°)[44–46]. It has been suggested that this difference is due to an inability of sparse stimuli to drive the edges of the RF above threshold. The size–tuning experiments use larger, steady-state drifting gratings that are matched to the optimal SF, so are more likely to drive the classical RF border and yield greater estimates of size.

The sizes computed in this study are best classified as an mrf. The median ($2\sigma = 0.58 \pm 0.12$) is a bit larger, but similar to, most of the previous mrf studies referenced above. Measuring the ssf with optimal sinewave gratings for all RFs in each two-photon field-of-view would be very time-consuming given the range of preferred orientations, SFs, and locations. Our random bar stimulus was designed to quickly yield quantitative measurements of complex and orientation selective V1 RFs for all cells in an imaging region that span multiple octaves of $f_o$. The width of the bars (0.2°) was chosen to (1) drive the majority of SF tuning selectivity in a parafoveal V1 population while (2) limiting the "smearing" that inflates the size estimate. We chose a relatively narrow bar width that drives the high side of SF sensitivity in the population in order to minimize the smearing of our SF envelope (see Supplementary Section VI, "Correcting for the effects of eye movements and stimulus bar width on RF size"). It should be noted that a subspace reverse correlation method, such as the one used in ref. [32], would overcome the challenging tradeoff of minimizing experiment time while matching $f_o$ and RF locations for a diverse population. However, this reconstruction method requires that the neurons have strong phase modulation. When attempted with our grating stimulus it yielded noisy RFs because they are mostly complex.

To our knowledge, only one study has directly compared $f_o$ to ssf[49]. They reported a trend that is partly consistent with Fig. 3a—size and $f_o$ do not scale—but their data are much closer to scale invariance. The cause of this discrepancy is probably due to differences in how the data were collected. For one, their population is likely pooled from a wider range of eccentricity, where scale invariance is going to hold more strongly. Also, their study used multi-unit activity, so did not distinguish between simple and complex cells. The few studies that compared RF size (as mrf) to $f_o$, for both simple and complex populations, found that this specific violation of scale invariance was largely limited to complex cells[10,17,18]. Our analyses show that pooled scale invariance can account for these observations (Figs. 3 and 4).

**Scale invariant pooling at other eccentricities**. The recordings from this study were limited to a very narrow range of eccentricity, which is both an advantage and pitfall. The advantage is that our trends are independent of the more global changes in RF scaling that correlate with eccentricity, unlike most other studies that examined scale invariance from a substantial population of electrode recordings. However, since we were limited to 2° to 4° from the fovea, it is unclear how our parameter estimates might generalize to other eccentricities. The pooled scale invariance model nonetheless allows for some predictions. In comparing RF size to $f_o$ in Fig. 3, we estimated a pooling window of 0.24° in visual space, which would be expected to scale inversely with magnification factor (mm/°) for a constant window in cortical space (mm). Using our retinotopy maps, the estimate of magnification was 2 mm/°, which normalizes the pooling window to 480 µm ($1\sigma$) of cortical distance. We may look to cortical distance as the most effective normalizer of the pooling window since the millimeter scale of the functional and anatomical architecture remains roughly constant across V1[50–52].

**Correcting for the effects of eye movements and finite stimulus bar width on RF size**. Two-photon imaging in the anesthetized-paralyzed primate is a valuable preparation for precise measurements of RF tiling in V1, since small eye movements in the awake preparation can easily corrupt measures of the fine structure in V1 RFs. However, even the anesthetized prep is susceptible to small drifts in eye position. In particular, we must consider the possibility that eye movements smear the measured RF envelope, making it seem wider along the axis of eye movement. Also, eye movements will make simple cells appear more complex, in a $f_o$-dependent manner—the phase modulation at higher $f_o$ will appear to be shallower. In addition to (potential) eye movement, the finite bar width (0.2°) of the random bar stimulus will inflate the measured RF size. In Supplementary Section VI, we measured the variance of slow eye movements, and added this to the smearing induced by the stimulus bar in order to identify an upper limit on the RF size inflation artifact. In short, we conclude that there is near-zero detectable correction to the fit relating RF size to $f_o$ in Fig. 3a, and $F1/F0$ to $f_o$ in Fig. 4a.

**Fluorescence nonlinearities**. The observed fluorescence signal in calcium imaging has a nonlinear relation to spike rates, which will bias tuning metrics. GCaMP nonlinearities are generally accelerating[25,53], which will have the effect of artificially narrowing a tuning curve or RF, in most operating regimes. However, our use of reverse correlation, which updates the stimulus multiple times within the integration window of the calcium signal's impulse response, can help to circumvent a static output nonlinearity[54,55]. Still, we did not directly compare to spike rates, so we cannot quantify the magnitude of the bias. Also, we were not able to reconstruct spike times in this preparation, as it is rather difficult in primate V1 for two main reasons. The first is that spike rates are high, which exposes a wider range of the nonlinearity for each cell, making deconvolution more challenging. The second is that we must account for breathing and heartbeat artifacts (see "Methods"). Although these issues are not insurmountable, they can make the spike reconstruction much less robust and ultimately yield noisier tuning functions than simply computing the stimulus-triggered average fluorescence. The potential effects of fluorescence nonlinearities on our main results are formalized with a simulation in Supplementary Sections IV.1 and IV.2 (Fig. S5), which draw two main conclusions. First, if the underlying spike rates yield scale invariant tuning, so would the fluorescence measurements. Second, if one were to correct for a GCaMP nonlinearity, the data would deviate even

further from scale invariance, and yield a larger integration window in both dimensions (° and cyc/°) of the pooling model.

**Scale invariance yields a variant V1 population envelope.** Due to V1 having small RFs and retinotopy, a small object in visual space (°) will elicit a localized response in cortical space (mm). If the object moves closer, it will elicit a broader response in the cortex. Similarly, if the object stretches along one axis, the response envelope in cortex will change along the corresponding axis. Studies suggest that the V1 response envelope is used by downstream areas to decode the shape and size of a stimulus[56–58]. However, in a scale invariant and retinotopically precise V1 population, the response envelope is also altered by the SF content of an object. For example, in showing two Gabor's with the same envelope but different SF carrier, the Gabor with lower SF will produce a wider response. It is possible that this could produce a perceptual confound if downstream areas are using the population envelope to decode the size and shape of objects. Given the results from previous studies[56,57], homogenizing scale invariant RFs into energy detectors of invariant size may be an important computation for downstream areas to decode object shape.

**Toward a general model of V1 tuning.** To build a general model of RFs in output layers of V1, future experiments will need to probe critical stimulus dimensions that were not measured here. For instance, binocular disparity arises within V1 and may also be linked to the maps of preferred SF, as suggested by the alignment between maps of $f_o$ and binocularity[33]. Furthermore, disparity maps have been shown in the cat with 2p imaging[59], thus lending support to their existence in the primate. Color is yet another complex stimulus dimension requiring additional investigation. As noted above, several studies have linked tuning of color and form, which suggests that color tuning can be connected to pooled scale invariance. However, to unify the relationship between color and the spatial parameters measured here, new measurements and modeling are required that compare them directly. Finally, V1 neurons exhibit several nonlinear tuning properties within and outside the classical RF. It may be that these nonlinear interactions are dependent on placement within the neighborhood of the functional architecture, combined with a spatially invariant pooling mechanism similar to what we have outlined here.

## Methods

**Animal preparation and surgery.** All procedures were conducted in accordance with guidelines of the US National Institutes of Health and were approved by the Institutional Animal Care and Use Committee at the University of Texas at Austin, which maintains AAALAC accreditation. We used two adult male rhesus monkeys (*Macaca mulata*), ages 6 and 13 years old. One of the two animals had been previously used for studies employing widefield imaging in the awake preparation. In this case, the widefield chamber was removed and replaced with a recording chamber and window used for an acute two-photon imaging session. The details of the acute two-photon imaging chamber are described in ref. [12]. In summary, a glass coverslip was pressed against the brain using small stabilization feet that were anchored to a surrounding titanium disk. The second animal was initially implanted with a chronic two-photon imaging chamber design, illustrated in Fig. S9.

Details of the method for injecting virus (rAAV:CaMKII-GCaMP6f) in both of the animals have been described previously[27]. In summary, a glass pipette (15 μm tip diameter) was first lowered through an opening in the imaging chamber, puncturing the pia. Injections were made at depths of 1.5, 1.0, and 0.5 mm using a Nanoject II. At each depth, 0.5 μL was delivered manually in 10 × 50 nL steps, with approximately 30 s pauses between steps.

On the day of recording they were anesthetized with ketamine (10 mg/kg, i.m.) and pretreated with atropine (0.04 mg/kg, i.m.). They were placed in a stereotaxic apparatus, in which the animal's head was rigidly held in the stereotaxic frame by ear bars, eye bars, and a palate clamp (David Kopf Instruments). Anesthesia was maintained throughout the experiment with sufentanil citrate (4–10 μg/kg/h, i.v.), paralyzed using pancuronium or vecuronium bromide (0.1–0.2 mg kg$^{-1}$ h$^{-1}$, i.v.),

and artificially ventilated using a small animal ventilator (Ugo Basile). The EKG, EEG, SpO2, EtCO2, heart rate, and body temperature were monitored continuously to judge the animal's health and maintain proper anesthesia. Dexamethasone (0.1 mg/kg, i.m.) and cefazolin (25 mg/kg, i.v.) were administered at the beginning to reduce brain swelling and prevent infections. We administered topical 1% tropicamide to dilate the eyes, along with non-refractive contact lenses to prevent drying.

The first animal underwent a 5-day acute procedure (1 ROI), and the second underwent repeated anesthetized recordings (2 ROIs). In the latter case, each recording session lasted for a total duration of up to 8 h from initiation to termination of paralysis. After termination of paralysis and injectable anesthesia, the animal was given a dose of atropine (0.1 mg/kg i.v.), neostigmine (0.1 mg/kg i.v.) and naloxone (0.04 mg/kg i.v.) to facilitate recovery[60]. A peripheral nerve stimulator was used to indicate induction and recovery from paralysis.

**Two-photon microscope setup.** Images were collected with a resonant scanning two-photon microscope from Neurolabware, together with acquisition software by Scanbox. The scan rate was set to a 15 Hz frame rate. We used a Chameleon Ultra laser set to 920 nm. The beam size was adjusted to slightly overfill the back aperture of the ×16, 0.8 NA objective (Nikon).

**Visual stimuli.** Visual stimuli were generated using the Psychophysics Toolbox extensions for Matlab[61,62] on a 17-inch CRT monitor (1024 × 768) with a refresh rate of 60 Hz. The monitor was gamma corrected using a Photo Research-655 spectroradiometer. The midpoint of luminance (gray level) was 55 cd/m$^2$. To focus the contralateral eye on the screen, V1 SF tuning curves were measured after placing a range of corrective lenses in front of the eyes. To identify approximate RF locations, we identified visible increases in fluorescence while "hand mapping" with a bar on the screen. Following this manual mapping, approximate RF locations were confirmed from a trial-based coarse retinotopy experiment. In short, a drifting grating inside a square (1° × 1°) aperture was shown at a different location on each trial, which gave a spatial tuning curve along the horizontal and vertical dimension of the monitor. Next, we proceeded to present the stimuli used to quantify RF properties.

The random grating stimulus (Fig. 2c) varied over eight orientations (Δ22.5°), four spatial phases (Δ90°), and seven logarithmically spaced SFs between 0.25 and 8.0 cyc/°, giving 224 possible gratings in the ensemble. They were all at max contrast. The gratings were played in direct succession of each other and updated every 133 ms. They were inside a 3° square aperture, with a gray surround. The surround and grating midpoint had a luminance of 55 cd/m$^2$.

The random bar stimulus (Fig. 2j) varied in orientation, position, and luminance. The bars were 2°–4° long (varied across ROI) and 0.2° wide. Orientation was sampled 10° apart and bar positions were sampled 0.1° apart. However, the tuning curves were binned by 2× before Gaussian fitting to compute parameters. For example, the tuning curves shown in Fig. 2n, o have orientation and position sampled at intervals of 20° and 0.2°, respectively. The two luminance levels of the bars were presented at the monitor's maximum and minimum. All other pixels were set to the midpoint luminance of 55 cd/m$^2$. For both random grating and random bar stimuli, there was approximately 20 min of total stimulus time, which was divided up into 20–40 blocks, with 5 s of gray screen between blocks.

**Model of scale invariance.** Here, we describe how the scale invariance model equations were derived. In each equation, a different RF parameter can be predicted from the preferred SF, $f_o$ (cyc/°). We begin with the scale invariance constraint on RF width

$$\sigma_{x,si}(f_o) = 1/\alpha f_o; \text{RF width (Fig. 2a, blue).} \tag{1}$$

We estimate that $\alpha = \pi$ based on previous studies of primate and cat V1 simple cells[18,28–30], and implies 2–3 ON–OFF subfields. Other parameters that depend on $f_o$ can be derived from Eq. (1) using the Fourier transform identity of a Gaussian. The Fourier transform of a Gaussian in the spatial domain ($x$) produces a Gaussian in the SF domain ($f$) where the standard deviations are related by $\sigma_x = (2\pi\sigma_f)^{-1}$. This can be plugged into Eq. (1) to predict SF bandwidth from $f_o$ for a linear RF.

$$\sigma_{f,si}(f_o) = \alpha f_o/2\pi; \text{linear SF bandwidth (Fig. 4a, blue).} \tag{2}$$

For logarithmic SF, we start with the following function that depends on linear SF bandwidth and $f_o$, which reduces to a constant under scale invariance:

$$\sigma_{\log(f),si}(f_o) = \log_2[(f_o + \sigma_{f,si})/f_o] = \log_2(1 + \alpha/2\pi) \sim 0.71 \text{ oct;} \\ \log \text{SF bandwidth (Fig. S3a, blue)} \tag{3}$$

Note that logarithmic bandwidth can instead be defined using the ratio between high- and low-pass cutoff of the SF tuning curve, as in previous studies[21]. However, we opted to define it as a function of $f_o$ and linear bandwidth ($\sigma_{f,si}$) to allow for analytical transformation between linear and logarithmic bandwidths in the output of the pooling model. For context, our definition (Eq. (3)) yields a value that is about 1 octave lower than using the more classically defined ratio between high- and low-cutoff.

Finally, we also derive orientation bandwidth from Eq. (1). Orientation bandwidth, $\sigma_\theta$, can be computed as $\mathrm{atan}(\sigma_{f,\mathrm{si}}/f_o/A)$, where $A = \sigma_y(f_o)/\sigma_x(f_o)$; i.e. "$A$" is the aspect ratio of the RF where $\sigma_y(f_o)$ is the RF length along the axis that is parallel to the preferred orientation, and $\sigma_x(f_o)$ is the width defined in Eq. (1). We used $A = 2$ (refs. [28–30]). This reduces to

$$\sigma_{\theta,\mathrm{si}}(f_o) = \mathrm{atan}\left(\frac{\alpha}{2\pi A}\right) \sim 14°; \text{ orientation bandwidth (Fig. 6a, blue).} \quad (4)$$

**Model of pooled scale invariance.** Here, we summarize all the equations and fits to the pooled scale invariance model. First, the pooling model in the spatial domain is the convolution between the scale invariant RF envelope and a weighting function (pooling window), which have widths defined as $\sigma_{x,\mathrm{si}}$ and $\sigma_{h(x)}$, respectively. Since variances of the convolved functions add, we can model the RF widths at the output of linear pooling as

$$\sigma_{x,p}^2 = \sigma_{x,\mathrm{si}}^2(f_o) + \sigma_{h(x)}^2 \text{ (Fig. 3a, green).} \quad (5)$$

The known variables from the data are $\sigma_{x,p}$ and $f_o$, which allowed us to estimate $\sigma_{h(x)} \sim 0.24°$ by taking the root-median of $\sigma_{x,p}^2 - \sigma_{x,\mathrm{si}}^2(f_o)$. The Eq. (5) fit is shown in Fig. 3a (green line), which is $\sqrt{(\pi f_o)^{-2} + 0.24^2}$.

We then used $\sigma_{h(x)}$ to help predict the phase modulation at the output of the pooling model, which is derived in the next section and given as

$$F1(f_o)/F0 = \pi * \exp[-(\sigma_{h(x)}2\pi f_o D)^2/2] \text{ (Fig. 4a, green),} \quad (6)$$

where $D$ is a free parameter and estimated to be 0.54.

The results also describe the pooling model in the SF domain, which is a convolution between the scale invariant Gaussian SF bandwidths and a Gaussian pooling window, defined as $\sigma_{f,\mathrm{si}}$ and $\sigma_{h(f)}$, respectively. The pooling model in the SF domain is therefore described as

$$\sigma_{f,p}^2 = \sigma_{f,\mathrm{si}}^2(f_o) + \sigma_{h(f)}^2 \text{ (Fig. 5a, green).} \quad (7)$$

Estimation of the one unknown coefficient, $\sigma_{h(f)}$, gives the plot in Fig. 5a (green line), which is $\sqrt{(f_o/2)^2 + 0.85^2}$.

Next, we showed that the pooling model can account for the logarithmic bandwidth, which we defined as

$$\sigma_{\log(f),p}(f_o) = \log_2[(f_o + \sigma_{f,p}(f_o))/f_o] \text{ (Fig. S3a, green)} \quad (8)$$

$\sigma_{f,p}$ is the fit to Eq. (7). Lastly, we define orientation bandwidth as

$$\sigma_{\theta,p} = \mathrm{atan}\left[\frac{\sqrt{\sigma_{f,\mathrm{si}}^2(f_o)/A^2 + \sigma_{h(f)}^2}}{f_o}\right] \text{ (Fig. 6a, green),} \quad (9)$$

where "$A$" is the aspect ratio of the RF. The numerator inside the brackets is the RF bandwidth ($\sigma$) in the 2D Fourier domain along the axis that is perpendicular to the dimension of the SF tuning curve. That is, it is tangent to the arc of orientation.

**Formulating predictions of F1/F0 in the pooling model.** We measured phase selectivity at the preferred SF ($f_o$) as F1/F0. Specifically, we obtained the peak-to-peak amplitude from the Fourier transform of the phase tuning curve, and then divided by the mean over phase (i.e. F1/F0). To model F1/F0 as a function of $f_o$ in pooled scale invariance, we start with the assumption that the pooled population of scale invariant RFs are simple cells with a half-wave rectifying nonlinearity. Next, we inherit the same Gaussian pooling window that was fit in Eq. (5): $\sigma_{h(x)} = 0.24°$. As formalized below, the input from each pooled RF is phase-shifted and Gaussian-weighted according to $\sigma_{h(x)}$, then they are all summed to yield the output cell's phase tuning (Fig. 4). As the simplest case, we start with a constant relative phase, whereby the absolute phase shift is coupled to the retinotopic shift of the envelope; i.e., the RFs look identical, but are shifted by the retinotopy (Fig. 4c, left). In this case, absolute phase shift (°) equals $f_o$ times the retinotopic shift of the RF ("$x$"deg). To formulate F1/F0 as a function of $f_o$, we start with convolution between the spatial pooling function and a scale invariant Gabor.

$$G_x(0, \sigma_{h(x)}) \circledast \mathrm{Gabor}_x(f_o),$$

where $G_x(0, \sigma_{h(x)})$ is the spatial pooling function—it is a Gaussian in the spatial domain ($x$) with peak = 1, mean = 0 deg, and SD = $\sigma_{h(x)}$ deg. The right side of the convolution is a scale invariant Gabor with carrier frequency $f_o$. Importantly, the Gabor is normalized by the square root of its L2 norm, which ensures that the Fourier amplitude at $f_o$ (i.e. the F1) is constant prior to the convolution. We are interested in the Fourier amplitude following the above convolution, which is proportional to the product of two Gaussians in the frequency ($f$) domain:

$$G_f\left(0, (\sigma_{h(x)}2\pi)^{-1}\right) * G_f(f_o; f_o/2).$$

The left side of the product is the Fourier amplitude of the pooling function and the right side is the Fourier amplitude of the Gabor. Both are unity amplitude Gaussians with the mean and SD in parentheses, in units of cyc/°. Evaluating at $f_o$ gives

the following, which is proportional to the "F1":

$$\exp[-(\sigma_{h(x)}2\pi f_o)^2/2].$$

For a half-wave rectifying nonlinearity, the maximum F1/F0 is $\pi$, so we scale accordingly to get

$$F1(f_o)/F0 = \pi * \exp[-(\sigma_{h(x)}2\pi f_o)^2/2],$$

which is a Gaussian centered at 0 cyc/°, with an SD of $(\sigma_{h(x)}2\pi)^{-1} = 0.66$ cyc/°. This function is shown in Fig. 4a (bottom dashed green), which underestimates the population's phase selectivity at each $f_o$. The above equations assume that all the scale invariant inputs have the same relative phase (e.g. they all have even symmetry), which means that any change in the absolute phase is coupled to the RF envelope. For a more accurate model, we reduced the rate of the absolute phase progression with a coefficient, denoted $D$, in Eq. (6) below. $D$ was fit by minimizing the mean-squared error.

$$F1(f_o)/F0 = \pi * \exp[-(\sigma_{h(x)}2\pi f_o D)^2/2] \quad (6)$$

$D = 1$ and 0 translate to constant relative and absolute spatial phase, respectively (Fig. 4c, left and right). The fit yields $D = 0.54$ (Fig. 4c, middle), which slows the advance of the carrier frequency, $f_o$, relative to that of the RF envelope—that is, absolute phase changes along the cortical surface at about half the rate of the retinotopic gradient. Note that $\sigma_{h(x)}$ was previously constrained without $D$ in the pooling model to account for RF width.

**Quantifying model performance with cross-validation.** We compared error distributions between the two competing models—scale invariance and pooled scale invariance—in predictions of RF width and SF bandwidth. For each model and RF parameter, a prediction error was computed for every data point using leave-one-out cross-validation. This means that the model was fit anew to generate a prediction for every data point, where the predicted data point was excluded from the calculation of parameter fits. This generated a distribution of squared errors for each model, defined for each data point ("$i$") as $[\log_2(\sigma_{i,\mathrm{data}}) - \log_2(\sigma_{i,\mathrm{prediction}})]^2$, where $\sigma$ indicates either RF width or SF bandwidth. To identify a significant difference in the mean-squared error between models for each parameter, a paired $t$-test was used.

**Processing calcium signals to generate tuning curves.** For each neuron, the random grating stimulus yields the average timecourse in response to each combination of orientation, spatial frequency, and spatial phase, $R_{\mathrm{grat}}(\mathrm{ori},\mathrm{SF},\mathrm{phase},t)$. The random bar stimulus yields the average response to bar orientation, position, and luminance, $R_{\mathrm{bar}}(\mathrm{ori},\mathrm{pos},\mathrm{lum},t)$. To arrive at these response kernels from the raw movies required four general steps: (1) rigid alignment of the frames to correct for lateral brain movement, (2) neuron identification using local autocorrelation of the movies, (3) filtering to remove resonant peaks in the power spectrum from breathing and heart rate, and (4) computing the stimulus-triggered average response. The details of these steps are given below.

To account for lateral brain movement, we performed rigid alignment of each frame using the Scanbox offline software toolbox, which uses the cross-correlation between images on successive frames. From there, cells were identified using the local cross-correlation image, whereby the timecourse of each pixel was cross correlated with the weighted sum of its neighbors[63]. This is given as

$$\mathrm{LocalX}(x,y) = \sum_t F(x,y,t)[F(x,y,t) \circledast \mathrm{DoG}(x,y)], \quad (10)$$

where $F(x, y, t)$ are the fluorescence values at each pixel and timepoint, $\mathrm{DoG}(x, y)$ is a difference-of-Gaussian image, and $\circledast$ indicates convolution in $x$ and $y$. The DoG rewards pixels that are correlated in time with their immediate neighborhood, but also penalizes them if they are correlated in time with the broader surround. The central Gaussian of the DoG was near the size of a neuron body ($\sigma = 3\,\mu m$), whereas the outer "suppressive" Gaussian had $\sigma = 20\,\mu m$ to capture the surrounding neuropil. The integral of each Gaussian was normalized. Including the suppressive surround in the weighting function yields brighter and more localized puncta in the resultant image, $\mathrm{LocalX}(x, y)$ (Fig. 2b). That is, groups of pixels about the size of a cell body could be more clearly disambiguated from the surround, which is important for the subsequent manual selection of cells. The puncta were manually selected using a "point-and-click" GUI. After clicking on a location, a local threshold was applied to $\mathrm{LocalX}(x, y)$ in order to identify the cell ROI. The pixels inside the ROI were then averaged to give the timecourse of each cell. On each trial, the timecourse of each neuron was passed through a double notch filter that removed the breathing and heartbeat artifact. The algorithm searched for two resonant peaks in a "typical" frequency range that were >3 SD outside of the noise at nearby frequencies. The timecourse of each neuron was Z-scored within each trial to account for minor changes in depth or fluorescence over the course of the experiment. From there, we computed the stimulus-triggered average, for 1 s after the stimulus onset, for all stimulus combinations, which are defined as $R_{\mathrm{grat}}(\mathrm{ori},\mathrm{SF},\mathrm{phase},t)$ and $R_{\mathrm{bar}}(\mathrm{ori},\mathrm{pos},\mathrm{lum},t)$ for the random grating (Fig. 2c) and random bar experiments (Fig. 2j), respectively. Next, we computed a tuning curve along each dimension of $R_{\mathrm{bar}}$ and $R_{\mathrm{grat}}$. The first step was to remove the time dimension, which entailed smoothing along time with a Gaussian ($\sigma = 50$ ms), and then taking the slice of the smoothed kernel at the peak response ($t_{\mathrm{optimal}}$).

To compute orientation and SF tuning curves from $R_{grat}$(ori,SF,phase, $t_{optimal}$), we averaged over phase. The 1D orientation tuning curve was computed as a weighted average over the SF dimension, where the SF weighting function was calculated as the average of $R_{grat}$(ori,SF) over ori. The 1D SF tuning curve was computed analogously. The orientation tuning curve was fit with a four-parameter Gaussian after a circular shift to center the peak. The orientation tuning width was calculated as the half-width of the Gaussian fit at 61% of the amplitude. In most cases, this value was equivalent to $\sigma$ in the Gaussian fit. However, when the fit yielded particularly large $\sigma$ (e.g. >90°), they differed such that using the half-width metric provided a better representation of bandwidth. The SF tuning curve was fit with a five-parameter difference-of-Gaussian (DoG) function, $A_1 \exp\left[-(f - \mu_1)^2/(2\sigma_1^2)\right] - A_2 \exp\left[-f^2/(2\sigma_2^2)\right]$. For linear SF bandwidth, we used the fit to calculate half-width at 61% of the peak, defined as $\sigma_f = (f_{hi} - f_{low})/2$. All the cells included in the analysis were "bandpass" (see data yield description below); however, 5% of these bandpass cells did not have a low-pass cutoff based on the 61% criterion, in which case we used the lowest SF (0.25 cyc/°) for $f_{low}$. Logarithmic bandwidth (Fig. S3) was computed from linear bandwidths and $f_o$ as shown in Eq. (3). Phase tuning was measured at the orientation and SF that was nearest to the peak in the fits. A sine wave was fit to the phase tuning curve, which gave $F1$ (peak-to-peak amplitude) and $F0$ (mean).

To compute RF width from $R_{bar}$(ori,pos,lum,$t_{optimal}$), we first averaged over the two luminances (black and white). Next, the optimal orientation was found after averaging over position. Finally, a Gaussian fit to the position tuning curve, at the slice of optimal orientation (i.e. "line weighting function"), gave the RF width (Fig. 3a, left). The details of calculating the x/y RF location (Fig. S2) are described in ref. [33]. In summary, this entailed fitting a 2D Gaussian to the inverse Radon transform of the response as a function of orientation and position.

To compute the normalized ON–OFF separation from the random bar stimulus (Tables 1, 2), we extracted the position tuning curve for the ON and OFF responses, at a single optimal orientation (Fig. 2o). The optimal orientation was based on the average ON + OFF response. A 1D Gaussian was fit to both the ON and OFF line weighting function. The normalized ON–OFF separation was then computed as $|\mu_{ON} - \mu_{OFF}|/[\sigma_{ON} + \sigma_{OFF}]$. Unlike other metrics, ON–OFF separation required that we compute both the ON and OFF subfield, as opposed to the average. Several cells had weak responses to the ON bars. We nonetheless kept the data yield constraints (below) the same, which were based on the fits to the ON + OFF response.

The data yield in each analysis was defined as follows. Neurons in the ROI were first identified from manual selection in the local cross-correlation image (Fig. 1b). The local cross-correlation image was generated during presentation of the random grating stimulus. This study pooled from three ROIs in two animals, which yielded 34 (0.5 × 0.35 mm), 98 (1.0 × 0.7 mm), and 71 (1.0 × 0.7 mm) neurons from this manual selection stage. The first and smaller ROI is shown in Fig. 1 ($n = 34$). Figures 3–6 contain the second ROI ($n = 98$), and Fig. 7 contains the third ROI ($n = 71$). After manual selection in the cross-correlation image, neurons were excluded from analyses of parameter comparison that did not yield accurate parameter fits. First, neurons were excluded from analysis of the random grating stimulus (Fig. 2c) that did not have orientation and SF tuning curves that could be fit with respective Gaussian and DoG curves that accounted for at least 70% of the variance (24/203 excluded). We did not further exclude neurons based on the phase tuning curve fit because this would have excluded flat tuning curves (i.e. complex cells). Next, neurons were excluded from analyses of the random bar stimulus (Fig. 2j) that did not yield a 2D Gaussian fit to the RF envelope that accounted for 70% of the variance (25/203 excluded). In the case of the RF width vs. $f_o$ analyses (Figs. 3a and 4b), we combined results from random grating and random bar stimuli, which combined the two exclusion criteria, resulting in 41/203 total exclusion.

Finally, we always excluded an additional five cells that were deemed "low pass" in their SF tuning, as they are undefined in a model of scale invariance. Although they are excluded from statistics, they are shown in scatter plots on the left side of the x-axis (Figs. 3a, 5a and 6a). Low pass was defined as having a fit to the SF tuning curve with a peak at the minimum SF.

**Quantifying functional organization**. To quantify the tuning similarity between neurons at varying distances, or "functional clustering", we computed joint statistics between cell pairs at discrete pairwise distances. For non-circular variables, we computed the Pearson correlation coefficient between the pairings at each distance. For orientation preference in Fig. S1c, we used a metric for circular variables used previously[12]. Specifically, we computed, $r = \mathrm{real}\langle e^{2i(ORI_i - ORI_j)}\rangle_{i,j}$, where $ORI_i$ and $ORI_j$ are the preferred orientations of neuron pairs $i$ and $j$, and the brackets indicate the mean over all pairs separated by a given cortical distance. $r$ will be 1 if $ORI_i$ and $ORI_j$ are always the same, −1 if they are 180° apart, and ~0 if they are 90° apart or independent. To compute the significance of orientation clustering at each distance, the neuron locations were randomly shuffled, and then clustering was recomputed, 1000 times. The $p$ value was the percentage of times that shuffling yielded stronger clustering.

In the case of parameters computed from the random grating experiment, the total number of unique pairs in the three ROIs was 5176, and divided up into bins of cortical distance at 75 μm intervals. The minimum number of pairs in a bin was

303 (0–75 μm) and the maximum was 892 pairs (300–375 μm). The total yield was different for the random grating and random bar experiments (see above). In the case of the random bar experiment (i.e. RF width, Fig. 3f), the total number of pairs was 4821, with highest and lowest bin-yields of 296 (0–75 μm) and 830 (375–450 μm).

**Reporting summary**. Further information on research design is available in the Nature Research Reporting Summary linked to this article.

## Data availability
The data from this study are available from the authors upon reasonable request.

## Code availability
The Matlab code that supports the findings of this study are available from the corresponding author upon reasonable request.

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

## Acknowledgements

This work was supported by grants from the University of Texas (BRAIN Seed Grant 365248) and NIH (U01NS099720).

## Author contributions

I.N. and E.S. designed the experiments. B.V.Z. created the viral construct. Y.C. and E.S. performed the surgical procedures. I.N. and H.K. collected the data. I.N. analyzed the data and wrote the paper. All authors assisted with revisions.

## Competing interests

The authors declare no competing interests.

## Additional information

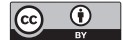

