## [Peer Review File · Nature Communications]

Reviewers' Comments:

Reviewer #1:

Remarks to the Author:

Chen and colleagues performed two-photon imaging in superficial layer of macaque primary visual cortex (V1) to examine local functional organization of monkey V1 and reported relationships between preferred spatial frequency (SF) and four other tuning properties: RF size, F1/F0, spatial frequency bandwidth, and orientation tuning bandwidth. They also demonstrated that these relationships were not accounted for by the scale invariance model. Then, they developed a new model, the pooled invariance model, that can describe the relationships between SF and other tuning properties. The pooled scale invariance model is a simple and unique model that accounts for several functional properties based on the preferred SF. However, there are several concerns about mapping of functional properties. Furthermore, all these functional organizations seem to be largely explained by well-known properties of cells in the blobs / interblob architecture in the superficial layer of macaque monkeys. The paper seems suitable for publication in more specialized journals after addressing the concerns below.

Major concerns:

- 1) The authors jumped into the characterization of tuning properties of neurons in Figure 1. However, it is important to confirm that how accurately these tuning properties are extracted from the calcium signals. Some figures in Supplementary Figure 1 should be demonstrated in the main figures, and more details should be described in the Methods. For example, how did the authors treat the problem of neuropil signal contamination? If there were some contamination of neuropil signal, it could broaden the RF size and other tuning properties, which could affect all the conclusions of this paper significantly. Further, if the estimation of the tuning properties were noisy due to low signal-to-noise ratio in some neurons, it could also broaden the tuning properties. If the authors select only neurons with high signal-to-noise ratio, would it affect the conclusions?
- 2) The authors reported several functional tuning properties. However, I am afraid that some tuning properties, such as RF size, may not be precisely evaluated to invalidate the scale invariant model. For example, accurate mapping of On and Off RFs is the most critical points of this paper to validate the pooled invariance model. The authors used bars with 0.2-degree width, bar positions were sampled 0.1 degree apart, and the tuning curves were binned by 2x before Gaussian fitting. If so, the spatial resolution of RF position and size was 0.2 degree. Then, it becomes hard to estimate the RF size smaller than 1/5 degree. However, the authors discuss RF size up to 1/16 degree in Figure 2A, 2B, which is beyond the spatial resolution of their RF size measurement. Indeed, the measured RF sizes are floored around 1/5 degree and I am afraid that this is due to their spatial resolution of the RF measurement. This would significantly affect the conclusions of this paper.
- 3) Past literature suggested that neurons in blob are less selective for orientation, prefer low spatial frequency, contain more simple cells. This seems to explain many findings in this paper: the authors found that low SF preferring cells had broader orientation tuning and higher F1/F0.
- 4) I also concern generality or reproducibility of the model predictions because the model was based on a relatively small number of imaging areas (three ROIs from two animals) and cortical eccentricity examined are limited, so the model could be overfitted to the used dataset. Presentation of more data from various eccentricity makes the results convincing. The authors should present results (e.g., scatter plots and functional and prediction maps) for individual ROIs and demonstrate reliability of model predictions independently in each ROI.

Minor points:

- 1) The preferred SF was negatively or positively correlated to other functional properties. In such a situation, if the model explains one variable well, the model prediction should be correlated to other variables. I am afraid that these relationships may cause overestimation of model predictions for

several properties. For example, in the $F1/F0$ map in the example case shown in Figure 6, the pooled scale invariance model incorrectly predicts a cluster of large $F1/F0$ value. Furthermore, the pooled model does not capture the orientation bandwidth specifically for the higher SF neurons as shown in Figure 5A, which may suggest that the pooled scale invariance model cannot be applied to the orientation bandwidth. I recommend that the author should precisely evaluate and describe limitations of the pooled scale invariance model.

One possibility of deviations described above is that other factors affect the functional properties independent of the preferred SF (or the pooled scale invariance model). For example, does an orientation map structure interact the pooled invariance model? Does the model predict the functional properties uniformly within an imaging area or predict well only in a specific part?

2) Pooled scale invariant model assumes convergent input from locally scale invariant population. Do the authors hypothesize that scale invariance holds in layer 4? More detailed description of scenario would be helpful.

3) 1. The idea of scale invariance (and why some tuning parameters can be calculated from SF) may be unfamiliar to researchers outside the field. It would be helpful to introduce the concept more in details. Further, it would be helpful if the 2-D gabor functions of scale invariant model and their Fourier transformations are described as formula in Methods.

4) In graphs that represent spatial clustering of functional properties (e.g., Figure 2C), statistically significant points are seemed to be represented by filled circles, while the legend describe that asterisk indicates significant point. Also, asterisks are difficult to see in some figures.

5) In Fig.1 the numbers of cells depicted seem different across panels: some cells are missing in some panels.

6) In Fig.1 and Fig.6, the sizes of cells are too large and they are overlapping. It would be better to reduce their size for visualization.

7) In the legend of Figure 3B, "we took the ratio following ratio" should be "we took the following ratio"?

8) In the line 2 of the section, "Formulating predictions of $F1/F0$ in the pooling model" in Methods, "Fourier transform of the of the ...". "of the" is duplicated.

Reviewer #2:

Remarks to the Author:

In this work the authors explore numerous tuning properties in primate V1 using wide-field 2-photon imaging. The authors reproduce previous findings that receptive field properties, such as size, are related to a neuron's preferred spatial frequency. Importantly, the authors demonstrate that these relationships are not linear and instead can be explained by a simple nonlinear transformation termed 'pooled spatial invariance'. This simple computational model is used to explain observed V1 activity within a cortical neighbourhood, demonstrating functional organization of phase selectivity, tuning bandwidth, and receptive field size.

While the scientific content of this work is compelling and of interest to the field, the paper is poorly written. Multiple sections are extremely dense, with the authors' reasoning difficult to follow. Other sections are highly speculative and not adequately cited. Furthermore, while the figures demonstrate that predictions of tuning properties via pooled scale invariance are qualitatively similar to what was measured across the population, a lack of quantitative metrics makes comparison to naive models

difficult. Finally, the authors do not discuss their pooled spatial invariance model in the context of a simple schematic or neural mechanism that could be implemented within a local cortical region.

As a result of these major issues we cannot recommend this paper for publication in its current form.

Major

Recordings are taken of GCaMP calcium fluorescence which, as described in the discussion, is known to have a nonlinear relationship to spiking activity. Therefore it seems possible that neural activity could be linear with respect to preferred spatial frequency, but it appears linear due to these f_1/f_0 nonlinearities. While this may not be the case, we believe it to be the authors duty to demonstrate these findings are not consistent with a simple nonlinear fluorescence explanation.

There are no example neurons shown besides the four curves that appear in Figure 1A. We feel it difficult to verify the quality of f_1/f_0 tuning maps that are generated from fits to single units without a clear demonstration of fit quality. We suggest that for a handful of example neurons the authors illustrate the measured size and spatial frequency tuning against the scale invariant model prediction and the prediction from the authors' pooled invariance model.

Page 1, Introduction, par. 1 - The claim of "modelling this global trend...inputs from the retina" needs to be cited.

Page 1, Introduction, par. 1 - It is unclear what is meant by "...local variability in SF preference is tied to the rest of the spatial RF, such as the case of scale invariance".

Page 2, Results, par.2 - This paragraph reads like a figure caption, and is difficult to follow. This paragraph and the figure 1 caption should be rewritten. Further, here is where f_0 should be explicitly defined - "preferred spatial frequency" or "optimal spatial frequency" is less ambiguous than "SF preference".

Page 4, par. 4 - Random bars are described as a bandlimited stimulus. Bars are *not* band limited, at least not by any standard definition.

Page 7, par 1 - The broadening of RF size is described as pooling, but it seems like simple blurring in the spatial domain, which could be explained by other factors. It is important to argue why a pooling model is appropriate here, what exactly is being pooled, and how. To that end, the authors should create a figure in which they schematize their model in both the Fourier and spatial domains.

Page 10, par 1 - Figure 6 should be combined with Figure 1, and a metric should be included to quantitatively demonstrate pooled scale invariance as a more appropriate model. Pooled SI, however, has multiple degrees of freedom and some statistical measure or cross validation is required to demonstrate that there has been no overfitting. It is not appropriate for this paper to justify its conclusions by asking a reader to perform a visual comparison.

Page 12, par 1 - The claimed 'hitherto disconnection' needs to be cited.

Minor

In the title and abstract the authors are advised to mention exactly what novel topographic maps have been found.

Throughout the paper it is at times unclear what is meant by spatial frequency preference - is it peak tuning, bandwidth, etc. It is necessary to be precise.

Page 3, par. 1 - The sentence "First, most measured RF are..." contains a typo. RF to RFs?

Page 4, par. 2 - It is unclear what is meant by the symbol \sim in describing the Gaussian pooling variable.

Page 4, par. 2 - Typo around "Multiplying ... by magnification factor".

Page 6, par. 1 - "...indicating that phase is more..." is phase here supposed to mean absolute phase?

Page 7, par 1 - The claim that your recordings are much broader than 2-to-3 subfields of preferred spatial frequency needs to be cited and clarified.

Page 7, par 2 - "Linear SF bandwidth" needs to be clearly defined.

Page 7, par 2 - The last two sentences of this paragraph seem unnecessarily confusing.

Page 7, par 3 - Retinotopy maps are referenced as 'above', please indicate exactly what is being referenced.

Page 8, par 1 - The pooling model is also used to explain orientation bandwidth. In this case, pooling occurs in the Fourier domain. We find it difficult to understand how pooling is to occur simultaneously in both the spatial and Fourier domains in a mechanistic sense. Readers should be given an intuition here.

Page 9, par 2 - The claim "...this relationship between maps of orientation bandwidth and f_0 falls in line with previous studies on tuning within V1 maps" should include citations.

Page 10, par 1 - Again, this paragraph has text which reads closer to a figure caption.

Page 11, Table 1 - In the table caption you refer to 'four columns'...do you mean 'five rows'?. This caption is extremely difficult to parse and should be clarified.

Page 12, par 2 - The section titled "The predictability of orientation..." is inaccurate as the authors are demonstrating the prediction of orientation bandwidth, not how predictable (predictability) orientation bandwidth is.

Page 13, par 2 - Only in this paragraph is the following important point reiterated: locally, there is a departure of scale invariance of RF size and spatial frequency tuning. This point should be made more clear earlier on in the exposition.

Page 14, par 3 - These concluding remarks seem unnecessary.

Reviewer #3:

Remarks to the Author:

In this study, the authors performed 2-photon calcium imaging using GCaMP6f in anesthetized macaque V1 to quantify the topographic organization of spatial frequency (SF) preference (f_0), receptive field (RF) width, SF bandwidth, orientation bandwidth, and phase selectivity. The authors asked the question of how local variability in SF preference is tied to the overall spatial RF. The neural data reported in this study are rich and hard to get using conventional methods. Applying rigorous quantitative analysis, the authors showed that a "pooled scale invariance" model that integrates over a population of scale-invariant RFs can better describe the imaging results than a "scale-invariant"

model. The authors have made several novel findings. For examples, they found a previously unidentified map of $F1/F0$, suggesting that simple cells (and complex cells) tend to cluster together; They found that orientation bandwidth and SF preference are negatively correlated, which is not predicted by the scale-invariant model; They further found that using a Gaussian weighting function that has the same $\sigma(h(f))$, they can predict both SF bandwidth and orientation bandwidth. Together, this study provides a compact, sensible explanation of the RF width and SF bandwidth from f_0 and uses f_0 to account for variability in orientation bandwidth and phase modulation.

1. In the method, it is described that there are a total of 224 possible gratings in the stimulus ensemble and each grating was shown every 133 ms. What is the inter-stimulus interval (ISI) between the presentations of two gratings? The temporal dynamics of calcium imaging are relatively slow (see Supplementary Figure 1E and L). Fast presentation of a sequence of gratings with brief ISI may cause calcium responses elicited by different stimuli to merge.

2. In the data describing the relationship between orientation bandwidth and SF preference, the pooled scale invariance model (the green line) fits the data reasonably well, except this model completely misses the data points below the blue line (prediction based on scale invariance model) at high SF preference. It appears that a linear model can fit the data well and perhaps better. What is the goodness of fit of a linear model for the data in Fig. 5A, in comparison to the pooled scale invariance model? If two models have different numbers of parameters, Akaike information criterion may be used for comparing the goodness of fit.

3. The data fit in Figure 2A is not well constrained at low SF preference, which appears to be critical to differentiate the pooled scale invariance model from, e.g. a linear model with a shallower slope relative to that of the scale invariance model. The lack of constraint at low SF preference also appears in Figs. 4A and 5A.

4. Besides the clustering of spatial tuning maps given the clustering of f_0 , does the pooled scale invariance model make other predictions that can be tested in the future study?

Minor

1. The manuscript is written concisely, which is a strength. However, I found in some places, the description is a bit dense, especially for a journal with broad readers. E.g., in the section "Phase selectivity as a function of spatial frequency preference" on page 5, first paragraph, it would be helpful to state briefly that simple cells tend to have greater $F1/F0$ than complex cells, which will make it easier for readers to follow. The authors may look through the manuscript to try to make the paper more accessible to broader readers.

2. Between Figure 2D and 2E, why the neurons do not match exactly? It looks like some neurons in D are not shown in E and vice versa.

3. Legend of Figure 3, row 4, Fig. 3A may be Fig. 2A.

4. Page 19, second paragraph, 3rd line: average over orientation or SF?

We would like to thank the reviewers for their positive comments regarding the scientific advance, along with their detailed and insightful criticisms. We addressed every comment, and we believe the manuscript is significantly improved as a result. Below, we first preview some changes that were prompted by more general concerns among reviewers.

- Scale invariance and pooled scale invariance are the two V1 models discussed throughout the manuscript. To help explain how these two models are related to each other and the data, we have added several explanatory figures throughout the manuscript. For instance, each main result figure (Figs. 3-6) has a panel devoted to illustrating how the model of pooled scale invariance is a hierarchical model, whereby scale invariant simple cells are at the input.
- Consistent with the point above, the text has been modified to be more amenable to a broad audience. For instance, the Results now begin with a description of the classic V1 model that precedes this study, scale invariance (Fig. 1).
- We validate our tuning measurements with two-photon imaging on several fronts. We now include many more examples of raw single cell tuning and corresponding fits in the main text and supplemental. Furthermore, we include an analysis of fluorescence nonlinearities and neuropil contamination in the supplemental.
- We have compared the performance of pooled scale invariance and scale invariance, using a cross-validation method. Pooled scale invariance is superior (Final results section).

Major concerns:

1) The authors jumped into the characterization of tuning properties of neurons in Figure 1. However, it is important to confirm that how accurately these tuning properties are extracted from the calcium signals. Some figures in Supplementary Figure 1 should be demonstrated in the main figures, and more details should be described in the Methods. For example, how did the authors treat the problem of neuropil signal contamination? If there were some contamination of neuropil signal, it could broaden the RF size and other tuning properties, which could affect all the conclusions of this paper significantly. Further, if the estimation of the tuning properties were noisy due to low signal-to-noise ratio in some neurons, it could also broaden the tuning properties. If the authors select only neurons with high signal-to-noise ratio, would it affect the conclusions?

Thank you for raising these concerns. In certain instances, we believe these issues were accounted for but not sufficiently described in the text, which has now been addressed. Supplemental Figure 1 has been transferred to the main text as Figure 2. Also, we have added new analyses and several new figures in the main text and Supplemental to further address these issues. These changes are summarized below:

NEUROPIIL CONTAMINATION

- 1) Our method of ROI selection picks neurons with activity that is more independent from the surrounding neuropil. We have included additional text in the Methods to help clarify. e.g. *“The Difference-of-Gaussian [weighting function] rewards pixels that are correlated in time with their immediate neighborhood [a cell body], but also penalizes them if they are correlated in time with the broader surround [the neuropil].”*

- 2) We have added a detailed analysis in the Supplemental on neuropil subtraction, and its effect on tuning metrics. See Supplemental Figure 7. In summary, this processing step has minimal effect on the width of tuning curves, showing that neuropil contamination is not a factor in our results. This same analysis was performed in a previous study (Nauhaus et al 2016; Supplemental) using the indicator OGB in the primate, which showed a very similar result.

SNR AND FITTING

- 1) Noise does not systematically broaden measured tuning, it only makes the measurement less accurate. We have performed a simulation here to show this. Each data point in the scatter is a simulated orientation tuning curve with a substantial amount of noise added (see example inset) Most importantly, the unity line passes through the center of the distribution, showing that the noise does not create a biased estimate of tuning width.

Noise does not systematically broaden measured tuning: A substantial amount of Gaussian noise was added to Gaussian tuning curves of variable width, and then a Gaussian was fit to the noisy data points using the same procedure as in the paper. In the scatter plot below, the x-axis is the “actual σ ” of the Gaussian to which noise was added, and the y-axis is the “measured σ ” after adding noise. The inset shows the noisy tuning curve and Gaussian fit for the red data point.

- 2) At the same time, the tuning curves included in our analysis are not noisy. This is the very criterion of our data yield: variance accounted for by the fit (see Methods). In Supplemental Figure 4, we have now included the raw tuning measurements and tuning fits from every neuron in one field-of-view.

Below are the position tuning curves, and Gaussian fits, of every neuron in one of the ROIs. In Supplemental Fig. 4, this panel is adjacent to the phase tuning, spatial frequency tuning, and orientation tuning. This supplemental figure also shows which cells did not pass the noise threshold.

Supplemental Figure 4A. All position tuning curves (black) and Gaussian fits (red) from one region-of-interest. Curves without a red fit were deemed too noisy for inclusion in further analysis.

NONLINEARITIES:

Related to the question of tuning measurements, calcium imaging has known nonlinearities in the transformation between spikes and fluorescence. In Supplemental Figure 5, we provide new analyses of how nonlinearity is expected to impact our results.

2) The authors reported several functional tuning properties. However, I am afraid that some tuning properties, such as RF size, may not be precisely evaluated to invalidate the scale invariant model. For example, accurate mapping of On and Off RFs is the most critical points of this paper to validate the pooled invariance model. The authors used bars with 0.2-degree width, bar positions were sampled 0.1 degree apart, and the tuning curves were binned by 2x before Gaussian fitting. If so, the spatial resolution of RF position and size was 0.2 degree. Then, it becomes hard to estimate the RF size smaller than 1/5 degree. However, the authors discuss RF size up to 1/16 degree in Figure 2A, 2B, which is beyond the spatial resolution of their RF size measurement. Indeed, the measured RF sizes are floored around 1/5 degree and I am afraid that this is due to their spatial resolution of the RF measurement. This would significantly affect the conclusions of this paper.

We understand how this can be counterintuitive, but we would like the opportunity to convince the reviewer with a simulation that this is not a problem. As we show below and in the Supplemental, the broadening artifacts created by the width of the bar are very small and not sufficient to alter the computed pooling model or overall conclusions. This was shown analytically in the Supplemental Material, but to help make this more concrete, here we provide a simulation that compares the “actual” RF size against the “measured” RF size. This simulation closely mimics the measurement and analysis in the paper, such as the sampling and binning that the reviewer references. Our hope is that this new simulation convinces the reviewer that the 0.2 deg bar width is not problematic. We would be happy to include the results of the simulation below in the Supplemental if the reviewer feels it is necessary.

In summary, we simulate a simple cell with an ON and OFF subfield separation. We then “measure” both ON and OFF subfields by convolving them with the 0.2 deg bar, then sample every 0.1 deg, and finally binning every two samples to obtain 0.2 deg/sample. Some of these steps may not be necessary to prove the point, but our goal was to make it as explicitly similar to our analyses as possible. For a summary conclusion of these simulated results, see panel ‘B’ below, which has the same axes as Fig. 3A of the paper.

Figure simulating the effects of 0.2 deg bar stimuli, and downsampling on the RF size. A) Each of the four columns correspond to a different spatial frequency preference, shown in the title. This values are in the range of the data. Top row shows the difference of Gaussian model with ON and OFF subfield. Solid line is the “actual” subfield. Dashed line is the is what can be measured with the 0.2 deg bars. i.e. it is the actual subfield convolved with the 0.2 deg bar. Bottom row shows the RF envelope; ON + OFF. Solid line is the actual envelope. Dots simulate our measured data points of the RF envelope; they are the measured envelope sampled at 0.1 deg, followed by binning every two samples to achieve 0.2 deg sample rate. Title indicates the measured RF width of the actual and measured RF width, respectively. B) RF width vs SF preference plot, like Fig. 3A in the manuscript. The effect of the stimulus and our analysis cannot account for the deviation from scale invariance in green.

Finally, we want to make sure it is clear that the bars were only used to measure RF size and position. The F1/F0 metric (Fig. 4) was measured using gratings, not the bars. Although F1/F0 could have instead been measured with the bars, this would have indeed created errors related to what the reviewer has mentioned.

3) Past literature suggested that neurons in blob are less selective for orientation, prefer low spatial frequency, contain more simple cells. This seems to explain many findings in this paper: the authors found that low SF preferring cells had broader orientation tuning and higher F1/F0.

We agree that the general direction of some trends (not all) can be inferred from prior studies. A substantial amount of text in the Introduction and Discussion is devoted to referencing this prior work. However, inference leaves us without quantification. For instance, in the case of F1/F0, one cannot use the literature to deduce a model of its dependency on spatial frequency preference, or its clustering along the cortical surface, as we have done here. More generally, prior studies that independently link functional properties to blobs are clearly valuable from many standpoints, but they leave us with a disconnected set of descriptive observations that cannot ultimately be used to model V1 output. Lastly, although the connection between blobs and functional maps is relevant to this study, it is tangential to the issue of scale invariance, which it is at the heart of the paper.

4) I also concern generality or reproducibility of the model predictions because the model was based on a relatively small number of imaging areas (three ROIs from two animals) and cortical eccentricity examined are limited, so the model could be overfitted to the used dataset. Presentation of more data from various eccentricity makes the results convincing. The authors should present results (e.g., scatter plots and functional and prediction maps) for individual ROIs and demonstrate reliability of model predictions independently in each ROI.

Thank you for this suggestion. Although we are unable to collect additional data beyond the two animals and 155 units that have been included, we have now changed the presentation and quantification to explicitly represent the data and model fits for each ROI independently. In

summary, results are consistent within each independent ROI. The changes are described in '1' and '2' below.

- 1) **Data presentation:** All scatter plots now distinguish each ROI using different symbols. The one in Fig. 5 (SF bandwidth vs. preferred SF) is shown below. The trend of the ROIs looks similar and independently yield similar model parameters. The same presentation strategy is used in other figures as well (Figs 3,4, and 6).

Also, the Results show all functional and prediction maps from 2-of-3 ROIs. All maps of one ROI are shown in Figs 3-6, and all maps of another ROI are in Fig. 7.

Figure 5: SF bandwidth as a function of SF preference. (A) The scatter plot compares SF preference to SF bandwidth in three ROIs, each indicated by a different symbol. To the left of the y-axis are SF tuning curves (black dots) and DoG fits (red) of four example cells in one ROI outlined in 'D' and 'E'. Blue line is the scale invariance prediction of bandwidth, $f_0/2$...

- 2) **Model fitting:** The pooling model has three parameters. For each of these parameters, we now report the independent estimates from each ROI. For example, the section of text with this reporting on RF width is given here. "...which allowed us to estimate $\sigma_{h(x)} = 0.24^\circ$ using the entire data set, giving the green line in Fig. 3A. This parameter estimate was similar when measured independently for the 3 ROIs: $\sigma_{h(x)} = [0.18^\circ \ 0.24^\circ \ 0.24^\circ]$."

Overall, we agree with the reviewer that additional eccentricities would have been useful. We make this point in the Discussion. However, we believe that these results are likely to generalize within parafoveal V1, given the agreement between subsets of our data. Thank you for raising this as we believe the new plots and analyses strengthen the conclusions.

Minor points:

- 1) The preferred SF was negatively or positively correlated to other functional properties. In such a situation, if the model explains one variable well, the model prediction should be correlated to other variables. I am afraid that these relationships may cause overestimation of model predictions for several properties. For example, in the F1/F0 map in the example case shown in Figure 6, the pooled scale invariance model incorrectly predicts a cluster of large F1/F0 value. Furthermore, the pooled model does not capture the orientation bandwidth specifically for the higher SF neurons as shown in Figure 5A, which may suggest that the pooled scale invariance model cannot be applied to the orientation bandwidth. I recommend that the author should precisely evaluate and describe limitations of the pooled scale invariance model.

Unfortunately, we don't understand the reviewer's statement about "overestimation of predictions", such as how this might be created by multi-variable dependencies. However, it seems that the reviewer's main point is that the model does not fully capture the data in specific instances, such as the narrowly tuned orientation at the highest SFs. We agree that pooled scale invariance does not account for every feature of the data. However, our main goal is to convey that scale invariance – which is often used in modeling studies - can be rejected, and pooled scale invariance is a simple model that is far better at accounting for the data.

We now provide performance measures comparing the model of pooled scale invariance against scale invariance. See final section of Results. To provide a fair comparison between models that have different parameters, we trained the models and predicted RF properties using separate data points. The model of pooled scale invariance is superior to scale invariance.

One possibility of deviations described above is that other factors affect the functional properties independent of the preferred SF (or the pooled scale invariance model). For example, does an orientation map structure interact the pooled invariance model? Does the model predict the functional properties uniformly within an imaging area or predict well only in a specific part?

Thank you. We do believe that the organization of functional maps may be an important constraint in modeling RFs, such as in the case of orientation bandwidth (e.g. Ikezoe et al 2013). For instance, using the measured functional architecture around each neuron may provide a better fit of its tuning. This is now mentioned in the Discussion as an explanation for noise in the scatter plots: "This may also be a source of noise in the use of f_o to predict orientation bandwidth in the scale invariant pooling model – there will be randomness in the alignment between local minima in the maps of f_o and regions of diverse orientation preference." We believe that this is a bit beyond the scope of the current study, yet an important future direction.

Lastly, we now provide an explanatory figure that helps to better illustrate a model of pooling within the orientation and spatial frequency maps in Fig. 6C (below). In fact, the scale invariant pooling model may be considered as an idealized model of pooling from the local, but with the advantage of a simple mathematical description to model a population within the hypercolumn.

Figure 6C: Simulation of the pooling model in the 2D spectral domain, at $f_o = 1.5$ c/deg and 6 c/deg. At bottom is the 2D Fourier plane, where the color represents orientation and f_o increases linearly away from the center. Each location corresponds to a different scale invariant, 2D Gaussian. Pooling closer to the origin (@ 1.5 c/deg) yields broader orientation tuning than pooling further away from the origin (@ 6 c/ deg), assuming an invariant pooling function ($\sigma_{h(x)} = 0.24$ deg). The simulation on top shows a sample of weighted and shifted orientation tuning curves (blue) from the two indicated pooling locations, along with their superposition (green).

2) Pooled scale invariant model assumes convergent input from locally scale invariant population. Do the authors hypothesize that scale invariance holds in layer 4? More detailed description of scenario would be helpful.

We do believe it is plausible that the pooled scale invariance model arises from a feedforward pooling of a more scale invariant population in layer 4. This idea is consistent with laminar recording in V1. However, laminar distinctions of tuning properties in V1 is a bit contentious. We have attempted to make these ideas more explicit within the Discussion.

3) 1. The idea of scale invariance (and why some tuning parameters can be calculated from SF) may be unfamiliar to researchers outside the field. It would be helpful to introduce the concept more in details. Further, it would be helpful if the 2-D gabor functions of scale invariant model and their Fourier transformations are described as formula in Methods.

Thank you. We agree that the manuscript was in need of additional explanations of scale invariance and the pooling model. We have gone to great lengths to improve it in this direction. The manuscript now starts with a figure (Figure 1 at right) that describes scale invariance in the spatial and Fourier domains. Furthermore, each of the main results figures now has a panel devoted to illustrating the pooling model via a simulation.

4) In graphs that represent spatial clustering of functional properties (e.g., Figure 2C), statistically significant points are seemed to be represented by filled circles, while the legend describe that asterisk indicates significant point. Also, asterisks are difficult to see in some figures.

Thank you. We have re-generated the figure to make the significance points more clear and consistent with the Legend.

5) In Fig.1 the numbers of cells depicted seem different across panels: some cells are missing in some panels.

Thank you for raising this. We agree that it is a point that needed to be more clearly highlighted in the text. The yield in each map was determined by the variance accounted for of the tuning fit (see Figure Legend 3 and 7, and “data yield” of the methods). We have opted to show how the yield is slightly different for each parameter within the maps. However, the comparison between maps of RF width and SF preference, which is shown in the scatter plot (now Fig. 3A), is the intersection of the yield in the two maps.

6) In Fig.1 and Fig.6, the sizes of cells are too large and they are overlapping. It would be better to reduce their size for visualization.

Thank you. We have reduced the size of the dots in these panels to ensure that they can all be clearly seen.

7) In the legend of Figure 3B, “we took the ratio following ratio” should be “we took the following ratio”?

Thank you

8) In the line 2 of the section, “Formulating predictions of $F1/F0$ in the pooling model” in Methods, “Fourier transform of the of the ...”. “of the” is duplicated.

Thank you

Reviewer #2 (Remarks to the Author):

In this work the authors explore numerous tuning properties in primate V1 using wide-field 2-photon imaging. The authors reproduce previous findings that receptive field properties, such as size, are related to a neuron’s preferred spatial frequency. Importantly, the authors demonstrate that these relationships are not linear and instead can be explained by a simple nonlinear transformation termed ‘pooled spatial invariance’. This simple computational model is used to explain observed V1 activity within a cortical neighbourhood, demonstrating functional organization of phase selectivity, tuning bandwidth, and receptive field size.

While the scientific content of this work is compelling and of interest to the field, the paper is poorly written. Multiple sections are extremely dense, with the authors’ reasoning difficult to follow. Other sections are highly speculative and not adequately cited. Furthermore, while the figures demonstrate that predictions of tuning properties via pooled scale invariance are qualitatively similar to what was measured across the population, a lack of quantitative metrics makes comparison to naive models difficult. Finally, the authors do not discuss their pooled spatial invariance model in the context of a simple schematic or neural mechanism that could be implemented within a local cortical region.

As a result of these major issues we cannot recommend this paper for publication in its current form.

Thank you for the positive comments about the scientific advance, along with the detailed criticisms. We have attempted to thoroughly address each comment, in addition to an overall improvement of the writing. We feel that the manuscript has been significantly improved as a result.

Major

Recordings are taken of GCaMP calcium fluorescence which, as described in the discussion, is known to have a nonlinear relationship to spiking activity. Therefore it seems possible that neural activity could be linear with respect to preferred spatial frequency, but it appears linear due to these $f1/f0$ nonlinearities. While this may not be the case, we believe it to be the authors

duty to demonstrate these findings are not consistent with a simple nonlinear fluorescence explanation.

Thank you. We strongly agree that these additional analyses are warranted. Some of the general effects of nonlinearities may be intuitive, yet they require a more formal characterization. For this new assessment, we asked two questions. 1) Could the responses of a scale invariant population that get passed through a fluorescence nonlinearity account for our scale invariant pooling model? 2) How might the inversion of a fluorescence nonlinearity affect our measurements – specifically, how might our fits to the pooling model be altered if we corrected for a nonlinearity in the spike-to-fluorescence transformation? We used a power law nonlinearity for computational convenience, and assumed that it is accelerating based on reports in the literature with GCaMP6. The results are shown in Supplemental Figure 5 (below). The results of question 1 (dashed blue) and 2 (dashed green) are summarized in the Discussion: “The potential effects of fluorescence nonlinearities on our main results are formalized with a simulation in Supplemental Sections IV.1 and IV.2 (Fig. S5), which draw two main conclusions. First, if the underlying spike rates yield scale invariant tuning, so would the fluorescence measurements. Second, if one were to correct for a GCaMP nonlinearity, the data would deviate even further from scale invariance, and yield a larger integration window in both dimensions (deg and cyc/deg) of the pooling model. “

Supplementary Figure 5: Modeling the effect of a fluorescence nonlinearity on main results. Each panel shows data (black) and models (solid blue and green lines) that were presented in the main text. Here, we also show the potential effect of a GCaMP6 nonlinearity on the measured tuning of a scale invariant population (dashed blue). We also asked how a GCaMP nonlinearity would affect the pooling model. (A) The same data and models are shown as in Figure 3A. The dashed blue line is the result of passing the scale invariant spatial tuning curves through an accelerating nonlinearity (viz. GCaMP fluorescence). The dashed green line is the result of re-fitting the pooling model after inverting an accelerating nonlinearity. (B,C,D) Same data and models are shown as in Figure 5A,4A,6A, respectively. The modeling of nonlinearities (dashed lines) was performed as in ‘A’.

There are no example neurons shown besides the four curves that appear in Figure 1A. We feel it difficult to verify the quality of f1/f0 tuning maps that are generated from fits to single units without a clear demonstration of fit quality. We suggest that for a handful of example neurons the authors illustrate the measured size and spatial frequency tuning against the scale invariant model prediction and the prediction from the authors' pooled invariance model.

Thank you. We have now included several more examples of tuning and fits throughout the main text.

In particular, there are now a total of fourteen examples of spatial phase tuning and fits in the main text (Figures 2, 4, and 7). Furthermore, Supplemental Figure 4 includes the tuning and fits of every cell (n=71) from one ROI, shown below:

Supplementary Figure 4B: Response to each phase of a grating, at the optimal orientation and spatial frequency. F1/F0 from sinewave fit is in each title. Curves without a red fit were deemed too noisy for inclusion in further analysis.

In addition to the fourteen examples of spatial phase tuning in the main text, there are also fourteen examples spatial frequency (Fig. 2, 5, 7) and position (Fig. 2, 3, 7) tuning.

The examples of position tuning in Figure 7 are shown below, which includes the reviewer's nice suggestion of overlaying the model fits.

Figure 7A: Black dots are data points, blue Gaussian is the scale invariance prediction, green Gaussian is the pooled scale invariance prediction, and gray Gaussian is the fit to the data.

Adjacent to figure

Page 1, Introduction, par. 1 - The claim of "modelling this global trend...inputs from the retina" needs to be cited.

Thank you for catching this oversight. We have included 3 citations to studies that directly relate V1 magnification factor to the receptive fields and anatomy of retinal ganglion cells.

Page 1, Introduction, par. 1 - It is unclear what is meant by "...local variability in SF preference is tied to the rest of the spatial RF, such as the case of scale invariance".

Thank you. We have revised this and preceding sentences to clarify. This specific point now reads, "The question remains as to whether V1 RFs scale in proportion to preferred SF, within the hypercolumn, thus predicting other locally periodic maps of RF size and SF bandwidth."

Page 2, Results, par.2 - This paragraph reads like a figure caption, and is difficult to follow. This paragraph and the figure 1 caption should be rewritten. Further, here is where f_0 should be explicitly defined - "preferred spatial frequency" or "optimal spatial frequency" is less ambiguous than "SF preference".

Thank you. This section has been re-written. For reference, this section of the results is now toward the end. The reason being that we have taken the reviewer's suggestion in a comment below to combine what was Figure 1 and Figure 6 into a single summary figure – now Figure 7.

We have fixed the ambiguity in defining f_0 . The beginning of the Results starts with a cartoon description of the pooling model (Fig. 1), and a description of variables. f_0 is defined as "preferred SF". Also, the term SF preference has been replaced with preferred SF throughout the manuscript.

Page 4, par. 4 - Random bars are described as a bandlimited stimulus. Bars are *not* band limited, at least not by any standard definition.

We have reworded this to be more accurate. "The rectangular bar in our visual stimulus is composed of Fourier energy that decays at higher spatial frequencies."

Page 7, par 1 - The broadening of RF size is described as pooling, but it seems like simple blurring in the spatial domain, which could be explained by other factors. It is important to argue why a pooling model is appropriate here, what exactly is being pooled, and how. To that end, the authors should create a figure in which they schematize their model in both the Fourier and spatial domains.

Thank you. We have significantly expanded the schematization of the pooling model throughout the paper. We believe that the previous schematization as insets within scatter plots was overly subtle, so things are greatly improved by this. Each of the main Results figures (Figs. 3-6) now contain an independent panel devoted to clarifying the model in the spatial (Figs 3,4) and Fourier domain (Figs. 5,6). For reference, two example schematics (Figs. 4C and 6C) and the relevant section of the figure legend are below.

Figure 4 Legend: ... (C) Illustration of the pooling model in the spatial domain at $f_0 = 2$ c/deg, for the three examples of phase alignment plotted in 'A' (see gray circles in 'A'). At bottom are the 1D Gabor functions from the model of scale invariance. They are shifted (deg) and weighted according to the Gaussian in the pooling model ($\sigma_{h(x)} = 0.24$ deg). In the left example, relative phase does not change, whereas in the right example absolute phase does not change. The top green curves are the superposition of scale invariant inputs, where constant absolute phase (right) yields the greatest modulation. (D) Maps of F1/F0. Bottom panel is all simple cells and is the input to the scale invariant pooling model. Top panel is the output of scale invariant pooling using the phase clustering shown by the solid green curve in 'A'.

Figure 6C: Simulation of the pooling model in the 2D spectral domain, at $f_0 = 1.5$ c/deg and 6 c/deg. At bottom is the 2D Fourier plane, where the color represents orientation and f_0 increases linearly away from the center. Each location corresponds to a different scale invariant, 2D Gaussian. Pooling closer to the origin (@ 1.5 c/deg) yields broader orientation tuning than pooling further away from the origin (@ 6 c/deg), assuming an invariant pooling function ($\sigma_{h(x)} = 0.24$ deg). The simulation on top shows a sample of weighted and shifted orientation tuning curves (blue) from the two indicated pooling locations, along with their superposition (green).

Page 10, par 1 - Figure 6 should be combined with Figure 1, and a metric should be included to quantitatively demonstrate pooled scale invariance as a more appropriate model. Pooled SI, however, has multiple degrees of freedom and some statistical measure or cross validation is required to demonstrate that there has been no overfitting. It is not appropriate for this paper to justify its conclusions by asking a reader to perform a visual comparison.

Thank you for these requests. We believe the paper is improved by addressing them.

Figure 1 and Figure 6 are combined into what is now Figure 7. Figure 7 has been formatted to give a clearer summary of how the two models (scale invariance and pooled scale invariance) are related to each other, and the data.

The final section of the Results, "Summary of pooled scale invariance model and its performance", now quantifies the performance of pooled scale invariance against scale invariance. To generate the distribution of errors for each model, we used "leave-one-out cross validation". Pooled scale invariance generates significantly less error than two versions of scale invariance. In the first case, we used scaling coefficients based on the literature. In the second case, we fit the scaling coefficients. Pooled scale invariance is statistically superior to scale invariance in both cases.

Page 12, par 1 - The claimed 'hitherto disconnection' needs to be cited.

We have added detail to this sentence to allow for concrete citation.

“Some of the results described here can be gleaned from previous studies that relate single cell tuning to the surrounding architecture ^{8,37-39}, along with studies showing deviation from scale invariance ^{10,17,18,21,22}, yet their hitherto disconnection made it difficult to provide a simple and holistic model of V1 output.”

Minor

Thank you very much for all these detailed comments. We have addressed all of them. A few responses are interleaved

In the title and abstract the authors are advised to mention exactly what novel topographic maps have been found.

Thank you. We have now included this in the abstract. We have opted to not include this in the title.

Throughout the paper it is at times unclear what is meant by spatial frequency preference - is it peak tuning, bandwidth, etc. It is necessary to be precise.

Page 3, par. 1 - The sentence “First, most measured RF are...” contains a typo. RF to RFs?

-

Page 4, par. 2 - It is unclear what is meant by the symbol ~ in describing the Gaussian pooling variable.

Page 4, par. 2 - Typo around “Multiplying ... by magnification factor”.

Page 6, par. 1 - “...indicating that phase is more...” is phase here supposed to mean absolute phase?

Yes, thank you for catching this.

Page 7, par 1 - The claim that your recordings are much broader than 2-to-3 subfields of preferred spatial frequency needs to be cited and clarified.

Page 7, par 2 - “Linear SF bandwidth” needs to be clearly defined.

Page 7, par 2 - The last two sentences of this paragraph seem unnecessarily confusing.

Page 7, par 3 - Retinotopy maps are referenced as ‘above’, please indicate exactly what is being referenced.

Page 8, par 1 - The pooling model is also used to explain orientation bandwidth. In this case, pooling occurs in the Fourier domain. We find it difficult to understand how pooling is to occur simultaneously in both the spatial and Fourier domains in a mechanistic sense. Readers should be given an intuition here.

Page 9, par 2 - The claim "...this relationship between maps of orientation bandwidth and f_0 falls in line with previous studies on tuning within V1 maps" should include citations.

Page 10, par 1 - Again, this paragraph has text which reads closer to a figure caption.

Page 11, Table 1 - In the table caption you refer to 'four columns'...do you mean 'five rows'? This caption is extremely difficult to parse and should be clarified.

Page 12, par 2 - The section titled "The predictability of orientation..." is inaccurate as the authors are demonstrating the prediction of orientation bandwidth, not how predictable (predictability) orientation bandwidth is.

Page 13, par 2 - Only in this paragraph is the following important point reiterated: locally, there is a departure of scale invariance of RF size and spatial frequency tuning. This point should be made more clear earlier on in the exposition.

Page 14, par 3 - These concluding remarks seem unnecessary.

We have re-written the final paragraph of the Discussion to include more meaningful text. It now contains future directions and is titled, "Toward a general model of V1 tuning".

Reviewer #3 (Remarks to the Author):

In this study, the authors performed 2-photon calcium imaging using GCaMP6f in anesthetized macaque V1 to quantify the topographic organization of spatial frequency (SF) preference (f_0), receptive field (RF) width, SF bandwidth, orientation bandwidth, and phase selectivity. The authors asked the question of how local variability in SF preference is tied to the overall spatial RF. The neural data reported in this study are rich and hard to get using conventional methods. Applying rigorous quantitative analysis, the authors showed that a "pooled scale invariance" model that integrates over a population of scale-invariant RFs can better describe the imaging results than a "scale-invariant" model. The authors have made several novel findings. For examples, they found a previously unidentified map of $F1/F_0$, suggesting that simple cells (and complex cells) tend to cluster together; They found that orientation bandwidth and SF preference are negatively correlated, which is not predicted by the scale-invariant model; They further found that using a Gaussian weighting function that has the same $\sigma(h(f))$, they can predict both SF bandwidth and orientation bandwidth. Together, this study provides a compact, sensible explanation of the RF width and SF bandwidth from f_0 and uses f_0 to account for variability in orientation bandwidth and phase modulation.

We are grateful for the reviewer's comments on the rigor and novelty of the study, along with their valuable criticisms. We have addressed each comment and believe that the paper has been improved as a result.

1. In the method, it is described that there are a total of 224 possible gratings in the stimulus ensemble and each grating was shown every 133 ms. What is the inter-stimulus interval (ISI) between the presentations of two gratings? The temporal dynamics of calcium imaging are relatively slow (see Supplementary Figure 1E and L). Fast presentation of a sequence of gratings with brief ISI may cause calcium responses elicited by different stimuli to merge.

The inter-stimulus interval was zero, which is now given in the Methods. Thank you for catching this. As an aside, we have also transferred the referenced supplemental figure into the main text. It is now Figure 2.

Indeed, calcium transients from successive stimuli are expected to merge. However, this generates the same expectation of tuning as a more sparsely spaced stimulus, if the merging of responses is linear. In the case of an output nonlinearity (e.g. a fluorescence nonlinearity), the merging is theoretically helpful to recover the preceding linear stage. This is formalized in the context of calcium imaging in (Nauhaus et al J. Neurophys 2012), which compared electrode recordings to calcium imaging data using very similar stimuli to what is used in the current study (e.g. see cascade model in Fig. 4 of this prior study). The Nauhaus et al 2012 study leans on the principles established by those that have used similar stimuli with spikes (e.g. Chichilnisky et al 2001; Ringach et al 1997). More generally, it should be noted that similar assumptions are made with the use of reverse correlation stimuli with ISI shorter than the response timecourse of spike rates. These studies are referenced in the Discussion under "Fluorescence nonlinearities".

2. In the data describing the relationship between orientation bandwidth and SF preference, the pooled scale invariance model (the green line) fits the data reasonably well, except this model completely misses the data points below the blue line (prediction based on scale invariance model) at high SF preference. It appears that a linear model can fit the data well and perhaps better. What is the goodness of fit of a linear model for the data in Fig. 5A, in comparison to the pooled scale invariance model? If two models have different numbers of parameters, Akaike information criterion may be used for comparing the goodness of fit.

Thank you. The slope and y-intercepts of a linear fit are given in Table 2. See Table 2, column 4 for orientation bandwidth. At right, we show the comparison between the linear fit (gray) and the scale invariant pooling model (green) of orientation bandwidth. Both have a tendency to overestimate bandwidth at higher preferred SF. As Table 2 shows, both models have a strong correlation with the data. The pooled scale invariance model uses fewer parameters than fitting lines - Specifically, pooled scale invariance uses a single parameter to fit both orientation bandwidth and SF bandwidth, and this one parameter provides an interpretation of how tuning is shaped by the architecture (see new

Figure 6C for a schematic). Linear fits obviously require 4 parameters to fit both orientation and SF bandwidth. We offer the linear parameters in Table 2 for two reasons. The first is that they highlight deviation from scale invariance, and the second is that we understand that some researchers may prefer them for a given application. However, we believe the units of the parameters are less meaningful and are purely descriptive.

In addition to the correlation coefficients and linear fit parameters offered in Table 2, we have taken additional steps to compare the performance of a linear fit with pooled scale invariance. Please see the Supplemental section V, “Performance comparison between pooled scale invariance and a descriptive linear model”. In summary, we compared mean-squared errors with a paired t-test, where each prediction was trained on a different portion of the data - specifically, “leave-one-out cross-validation”. As the reviewer astutely surmised, a linear model performs slightly better than pooled scale invariance. We now show that the distribution of MSE was significantly different between the linear fit and pooled scale invariance. However, the difference in performance was subtle compared to the null model of primary interest, scale invariance, described in the main Results. The additional parameters used for all 4 linear models (8) vs. pooled scale invariance (3) incurred very little penalization in performance since there were so many more data points than parameters. We found the same general result to be true with Akaike Information.

Again, we emphasize that the pooling model offers an intuitive amendment to the classic model of scale invariance. This amendment performs far superior, and uses parameters that can be interpreted in the context of the functional architecture.

This is not directly related to the reviewer’s comment, but we also want to note that we fixed two mistakes in the original submission’s figure on orientation bandwidth. Visibly, the differences after correction are subtle, but they do improve the performance of the pooled scale invariance model. The first is that orientation bandwidth was being computed at a value that is slightly larger than 1σ . Specifically, we were using a cutoff at $1/\sqrt{2}$ of the peak, not 0.605 (i.e. 1σ). This moved the data points of the scatter plot up, slightly. The second is that the formulation of pooled scale invariance for ori bandwidth was inaccurate – the inputs to the model did not have scale invariant aspect ratios of the RF envelope. This changed the green line slightly. This latter change can be seen as a change to Equation 9.

3. The data fit in Figure 2A is not well constrained at low SF preference, which appears to be critical to differentiate the pooled scale invariance model from, e.g. a linear model with a shallower slope relative to that of the scale invariance model. The lack of constraint at low SF preference also appears in Figs. 4A and 5A.

Thank you.

Table 2 includes the results of a linear fit: slope (row 4), intercept (row 5), correlation coefficients (row 2), and p-values (row 3). This Table also shows performance statistics for the model of pooled scale invariance - Like the linear fit, pooled scale invariance is significantly correlated with the data in the case of each RF property. The model of pooled scale invariance is more significantly correlated with the data than a linear fit in the case of RF width and SF bandwidth (See Table 2, row 3 vs. row 7).

Although this difference is subtle, it implies that the curvature in the pooled scale invariance model is actually reflected in the data in these cases.

In addition, we now provide additional quantification to Table 2 on the relative performance of a descriptive linear model for each of the figures that the reviewer mentions. As mentioned in the previous comment regarding orientation bandwidth, we make use of cross-validation to obtain model predictions, and then compare the distribution of errors. In summary, a fitted line in the log-log domain performs better ($p < 0.05$; t-test) than pooled scale invariance, yet requires additional parameters. Although linear fits require over twice as many parameters as pooled scale invariance, they incur very little penalization due to the relatively large number of data points. These results are given in Supplemental Section V. We provide these in the Supplemental to minimize complexity, as the comparison with the classic model of scale invariance is the most relevant point. At the same time, we would be willing to append the main Results text if the reviewers believes it would be helpful.

To be clear, we have chosen to highlight the pooling model over a descriptive model (such as a linear fit) for a couple important reasons that we have attempted to clarify in the text. For one, it builds on the classic model of scale invariant receptive fields with one extra linear step of a feedforward cascade. Second, each pooling parameter has meaningful units within the functional architecture. In summary, the pooling model will provide greater utility to ultimately model the cortex. To clarify this, we have now further explained the mechanistic interpretation of the pooling model with additional text and several new explanatory figures adjacent to each scatter plot.

4. Besides the clustering of spatial tuning maps given the clustering of f_0 , does the pooled scale invariance model make other predictions that can be tested in the future study?

This is a very important point. We do believe that it begs questions for future studies. We have included an additional paragraph in our concluding remarks of the Discussion, now titled "*Toward a general model of V1 tuning*". In summary, we discuss potential links to color tuning, binocular disparity, and tuning properties beyond the classical receptive field.

Another important future direction is accounting for more of the variance in the pooling model by using the measured tuning in the neighborhood around each neuron. This is now mentioned in the Discussion as an explanation for noise in the scatter plots: "*This may also be a source of noise in the use of f_0 to predict orientation bandwidth in the scale invariant pooling model – there will be randomness in the alignment between local minima in the maps of f_0 and regions of diverse orientation preference.*"

Minor

1. The manuscript is written concisely, which is a strength. However, I found in some places, the description is a bit dense, especially for a journal with broad readers. E.g., in the section “Phase selectivity as a function of spatial frequency preference” on page 5, first paragraph, it would be helpful to state briefly that simple cells tend to have greater $F1/F0$ than complex cells, which will make it easier for readers to follow. The authors may look through the manuscript to try to make the paper more accessible to broader readers.

Thank you. We have made the text easier to follow for a general audience. In particular, the Results now lead with a simple description of Gabor receptive fields, simple/complex cells, scale invariance, and how this all translates to the spectral domain. The new figure 1 is shown to the right as a reference. The title of the section is, “Defining the model of scale invariant V1 receptive fields”. The rest of the Results link back to this introductory figure and make a greater effort to explain some of the more specialized topics.

2. Between Figure 2D and 2E, why the neurons do not match exactly? It looks like some neurons in D are not shown in E and vice versa.

Thank you for raising this. We agree that it is a point that needed to be more clearly highlighted in the text. The yield in each map was determined by the variance accounted for of the tuning fit (see Figure Legends 3 and 7, and “data yield” of the methods). We have opted to show how the yield is slightly different for each parameter within the maps. However, the comparison between maps ultimately requires the intersection between the different parameters.

3. Legend of Figure 3, row 4, Fig. 3A may be Fig. 2A.

Thank you

4. Page 19, second paragraph, 3rd line: average over orientation or SF?

Thank you. It was written correctly, but not clearly. It has been re-written.

Reviewers' Comments:

Reviewer #1:

Remarks to the Author:

The manuscript is much more improved in this revision, although its novelty and significance are not clear yet. The relationships between preferred spatial frequency (SF) and four other tuning properties: RF size, F1/F0, spatial frequency bandwidth, and orientation tuning bandwidth seem to be largely explained by well-known properties of cells in the blobs / interblob architecture in the superficial layer of macaque monkeys.

Major point:

1. "We agree that the general direction of some trends (not all) can be inferred from prior studies. A substantial amount of text in the Introduction and Discussion is devoted to referencing this prior work. However, inference leaves us without quantification. For instance, in the case of F1/F0, one cannot use the literature to deduce a model of its dependency on spatial frequency preference, or its clustering along the cortical surface, as we have done here. More generally, prior studies that independently link functional properties to blobs are clearly valuable from many standpoints, but they leave us with a disconnected set of descriptive observations that cannot ultimately be used to model V1 output. Lastly, although the connection between blobs and functional maps is relevant to this study, it is tangential to the issue of scale invariance, which it is at the heart of the paper."

I agree that quantification is important. However, in the title the authors mention "novel topographic organization" and in the abstract the authors claim, "all of which were found to be topographically organized and correlate with preferred SF. Each of these newly characterized inter-map relationships." If these organizations are related to blob / interblob architecture, they are not really new, and the title and the abstract overemphasize the novelty of this study.

Minor

1. "We would be happy to include the results of the simulation below in the Supplemental if the reviewer feels it is necessary."

Yes, I think that it is necessary to include the simulation, because the authors' argument "Convolution of the rect function with a Gaussian produces a Gaussian-like function that has a variance that is the sum of the rect and Gaussian variances. This may be provable analytically" is not really proved in the supplementary material. Further, in our simulation (see review attachment), broadening seems slightly larger than their simulation.

2. Maps of orientation and preferred SF are displayed in Supplementary Figure S1. It would be useful to present these maps in the main figure in order to easily compare the SF map and other functional maps.

3. In the bottom panel of fig 3D, 4D, 5D, 6D, a circle and lines converge to a cell in the top panel. What does it mean?

4. In fig 3A and E, the cell numbers or positions of example cells seem to be incorrect. According to the tuning curves in fig 3A, RF width for the cell 3 looks narrowest, but color code in fig 3E indicates that the RF width of cell2 is smallest.

4 deg/c, $\sigma_{\text{on}} = \sigma_{\text{off}} = 0.0625$ deg

Reviewer #2:

Remarks to the Author:

In the revised manuscript the authors present a clear and concise description of the tuning properties of surface V1 neurons being predicted from the preferred spatial frequencies within a local population. The authors are to be commended for their thorough treatment of the reviewer's concerns, including the addition of multiple figures and supplementary analyses. Overall the document is well written, interesting, solid, and modestly speculative. We strongly recommend this article for publication.

Reviewer #3:

Remarks to the Author:

The authors have sufficiently addressed my concerns. The manuscript has improved significantly. I have no further comments.

We are sincerely grateful to all three reviewers for their time and dedication to this review. Our final responses to reviewer 1 are below.

Major point:

1. “We agree that the general direction of some trends (not all) can be inferred from prior studies. A substantial amount of text in the Introduction and Discussion is devoted to referencing this prior work. However, inference leaves us without quantification. For instance, in the case of F1/F0, one cannot use the literature to deduce a model of its dependency on spatial frequency preference, or its clustering along the cortical surface, as we have done here. More generally, prior studies that independently link functional properties to blobs are clearly valuable from many standpoints, but they leave us with a disconnected set of descriptive observations that cannot ultimately be used to model V1 output. Lastly, although the connection between blobs and functional maps is relevant to this study, it is tangential to the issue of scale invariance, which it is at the heart of the paper.”

I agree that quantification is important. However, in the title the authors mention “novel topographic organization” and in the abstract the authors claim, “all of which were found to be topographically organized and correlate with preferred SF. Each of these newly characterized inter-map relationships.” If these organizations are related to blob / interblob architecture, they are not really new, and the title and the abstract overemphasize the novelty of this study.

First, we want to reiterate that we greatly appreciate the reviewer’s dedication to helping us improve this manuscript.

In hopes of partly satisfying the reviewer on this point of novelty, we have altered some wording in the title and abstract. The title no longer has “novel” in front of “topographic organization”. In the abstract, we have removed “for the first time” in front of the statement regarding the combined characterization of these functional maps.

Minor

1. “We would be happy to include the results of the simulation below in the Supplemental if the reviewer feels it is necessary.”

Yes, I think that it is necessary to include the simulation, because the authors’ argument “Convolution of the rect function with a Gaussian produces a Gaussian-like function that has a variance that is the sum of the rect and Gaussian variances. This may be provable analytically” is not really proved in the supplementary material. Further, in our simulation (see review attachment), broadening seems slightly larger than their simulation.

This is now included in the Supplemental.

It is nice to have the simulation replicated, in spite of what may be a subtle discrepancy. We can, however, point to a source of the apparent discrepancy: In the reviewer's version of the simulation, they did not quite match the amplitude of the "measured" and "actual" subfields. Specifically, a Gaussian fit to their measured curve (blue/red points) will have slightly higher amplitude than the actual (blue/red solid lines) curve. This inflates the apparent difference in width between measured and actual.

2. Maps of orientation and preferred SF are displayed in Supplementary Figure S1. It would be useful to present these maps in the main figure in order to easily compare the SF map and other functional maps.

Thank you for the suggestion. We agree that it would be useful to include these maps for reference in the main text, yet it is challenging to integrate the additional panels without compromising the layout.

3. In the bottom panel of fig 3D, 4D, 5D, 6D, a circle and lines converge to a cell in the top panel. What does it mean?

It was used as an illustration to show that the bottom map is the scale invariance prediction, and the top map is the pooled scale invariance prediction. However, we decided that it was not very useful, so it has been removed.

4. In fig 3A and E, the cell numbers or positions of example cells seem to be incorrect. According to the tuning curves in fig 3A, RF width for the cell 3 looks narrowest, but color code in fig 3E indicates that the RF width of cell2 is smallest.

Thank you for catching this. They were in reverse order.

Reviewer #2 (Remarks to the Author):

In the revised manuscript the authors present a clear and concise description of the tuning properties of surface V1 neurons being predicted from the preferred spatial frequencies within a local population. The authors are to be commended for their thorough treatment of the reviewer's concerns, including the addition of multiple figures and supplementary analyses. Overall the document is well written, interesting, solid, and modestly speculative. We strongly recommend this article for publication.

Reviewer #3 (Remarks to the Author):

The authors have sufficiently addressed my concerns. The manuscript has improved significantly. I have no further comments.